# Text-to-Decision Agent: Offline Meta-Reinforcement Learning from Natural Language Supervision

**Shilin Zhang**[1]* **Zican Hu**[1]* **Wenhao Wu**[1]* **Xinyi Xie**[1] **Jianxiang Tang**[1]
**Chunlin Chen**[1] **Daoyi Dong**[2] **Yu Cheng**[3] **Zhenhong Sun**[45✉] **Zhi Wang**[1✉]

[1] Nanjing University [2] University of Technology Sydney [3] The Chinese University of Hong Kong
[4] Australian National University [5] University of New South Wales

{shilinzhang, zicanhu, wenhaowu, xinyixie, jianxiangtang}@smail.nju.edu.cn
{clchen,zhiwang}@nju.edu.cn daoyi.dong@uts.edu.au
chengyu@cse.cuhk.edu.hk zhenhong.sun@anu.edu.au

## Abstract

Offline meta-RL usually tackles generalization by inferring task beliefs from high-quality samples or warmup explorations. The restricted form limits their generality and usability since these supervision signals are expensive and even infeasible to acquire in advance for unseen tasks. Learning directly from the raw text about decision tasks is a promising alternative to leverage a much broader source of supervision. In the paper, we propose **T**ext-to-**D**ecision **A**gent (**T2DA**), a simple and scalable framework that supervises offline meta-RL with natural language. We first introduce a generalized world model to encode multi-task decision data into a dynamics-aware embedding space. Then, inspired by CLIP, we predict which textual description goes with which decision embedding, effectively bridging their semantic gap via contrastive language-decision pre-training and aligning the text embeddings to comprehend the environment dynamics. After training the text-conditioned generalist policy, the agent can directly realize zero-shot text-to-decision generation in response to language instructions. Comprehensive experiments on MuJoCo and Meta-World benchmarks show that T2DA facilitates high-capacity zero-shot generalization and outperforms various types of baselines. Our code is available at https://github.com/NJU-RL/T2DA.

## 1 Introduction

Reinforcement learning (RL) has emerged as an effective mechanism for training autonomous agents to perform complex tasks in interactive environments [1, 2], unleashing its potential across frontier problems including preference optimization [3], diffusion model training [4], and reasoning [5, 6] such as in OpenAI o1 [7] and DeepSeek-R1 [8]. Despite its achievements, one of the grand challenges is *generalization*: building a general-purpose agent capable of handling multiple tasks in response to diverse user commands [9]. Due to distribution shift and the lack of self-supervised pre-training techniques, RL agents typically struggle with poor generalization to unseen tasks [10]. Offline meta-RL tackles generalization via training on a distribution of offline tasks [11]. However, they usually rely on high-quality samples [12] or warmup explorations [13] to infer task beliefs for generalization at test time, while these supervision signals are expensive and even infeasible to acquire in advance for unseen tasks [14]. *This inspires us to explore a much broader source of supervision.*

Recently, large language models (LLMs), pre-trained on extensive text corpora, can encode a wealth of semantic knowledge with remarkable representation power and transferability [15]. The typical "text-

---

*Equal contributions. ✉Corresponding authors.

to-text" interface enables task-agnostic architectures to achieve broad generalization with minimal or no domain-specific data [16]. The availability of large-scale text collections for aggregate supervision during model pre-training has revolutionized the language and multi-modal communities in recent years [17]. Naturally, the above advancements highlight a promising question: *Could scalable pre-training methods that learn perception from the supervision embedded in natural language lead to a similar leap forward in developing generalist agents for offline meta-RL?*

RL typically learns from active interactions with the outer world tied to specific environment dynamics, with unique decision-level representation distinct from the unbounded perception-level representation of natural language [18]. Recent studies facilitate language grounding to decision domains from various aspects, such as LLMs as policies [19, 20, 21], LLMs as rewards [22, 23, 24], and language-conditioned policy learning [25, 26, 27]. Despite these efforts, how to harness unbounded representations of natural language knowledge for building generalist decision agents remains the following challenges: i) LLMs, trained on text corpora, typically lack grounding in the physical world and fail to capture any environment dynamics; ii) Leveraging LLMs for decision-making is prone to knowledge misalignment due to the semantic gap between the text and decision modalities; iii) Scalable implementations are necessary to fully harness language knowledge to train generalist decision models.

Figure 1: t-SNE visualization of Ant-Dir where tasks with target directions in $[0, 2\pi]$ are mapped to rainbow-colored points. Top: we encode multi-task data into dynamics-aware decision embeddings to capture task-specific environment dynamics. Bottom: we bridge the semantic gap between text and decision via contrastive pre-training. The aligned text embeddings follow a cyclic spectrum that exactly matches the periodicity of angular directions *in a physical sense*. This interesting finding shows that we effectively align text embeddings to comprehend environment dynamics and facilitate convincing language grounding in decision domains.

To tackle these challenges, we propose **T**ext-to-**D**ecision **A**gent (**T2DA**), a simple and scalable pre-training framework for offline meta-RL via aligning language knowledge with environment dynamics of decision tasks. First, we pre-train a generalized world model to encode multi-task data into dynamics-aware decision embeddings, effectively capturing task-specific environment dynamics. Second, inspired by CLIP [16], we predict which textual description goes with which decision embedding, efficiently bridging their semantic gap via contrastive language-decision pre-training. It distills the world model structure to the text modality and aligns text embeddings to comprehend the environment dynamics. Finally, we deploy T2DA on two mainstream conditional generation architectures, training generalist policies conditioned on aligned text embeddings. During evaluation, natural language is used to reference learned decision perceptions or describe new ones, and the agent can directly realize zero-shot text-to-decision generation according to textual instructions at hand. Extensive experiments show that T2DA transfers nontrivially to downstream tasks, achieves high-capacity zero-shot generalization, and outperforms various types of baselines.

In summary, our main contributions are as follows:

- We present a generalized world model designed to capture the environment dynamics, facilitating effective grounding of linguistic knowledge in decision domains.

- We introduce contrastive language-decision pre-training to bridge their semantic gap, aligning the text embeddings to comprehend the environment dynamics.

- We propose a simple and scalable framework that supervises offline meta-RL with natural language, and develop scalable implementations with the potential to train decision models at scale: Text-to-Decision Diffuser and Text-to-Decision Transformer.

## 2   Related Work

**Offline Meta-RL** learns adaptable policies from pre-collected datasets without requiring online environment interaction [11, 28]. It addresses two fundamental challenges: the distributional shift between behavior and learned policies in offline RL [29, 30], and the quick adaptation to unseen tasks with minimal data. Existing approaches can be broadly categorized into the memory-based (e.g., RL$^2$ [31] and LLIRL [32, 33]), the optimization-based (MAML [34] and MACAW [11]), and the context-based methods (e.g., PEARL [35], VariBAD [36], etc. [28, 13, 37]). Recent studies of in-context RL attempt to leverage the in-context learning capability of transformers to improve RL's generalization via casting RL as an across-episodic sequential prediction problem with few-shot prompting at test time [38, 39, 40, 41], such as AD [42] and DPT [43]. In general, the above approaches rely on high-quality samples or domain knowledge to infer task beliefs during evaluation, limiting their generality with this restricted form of supervision. This inspires us to explore a much broader source of supervision from natural language.

**LLMs for RL**. Existing studies on leveraging LLMs for RL domains can be categorized into three main threads: LLMs as rewards, LLMs as policies, and language-conditioned policy learning. The first kind uses LLMs or VLMs (vision-language models) to automate the generation and shaping of reward functions [44, 22, 23, 24] or state representations [45], providing denser information to facilitate RL training. The second kind utilizes LLMs or VLMs as the policy backbone [18, 46, 47, 48]. Vision-language-action (VLA) models provide a direct instantiation of using pre-trained VLMs for robotics, fine-tuning visually-conditioned VLMs to generate robot control actions, such as RT-2 [49], OpenVLA [50], and $\pi_{0.5}$ [51]. In contrast, we leverage LLMs to obtain task representations for offline meta-RL and train conventional RL policies based on decision transformer or diffuser architectures.

**Language-conditioned policy**. We focus on the third kind that learns a language-conditioned policy, extracting the world knowledge encoded in LLMs to help train offline meta-RL agents. Previous studies acquire language embeddings using pre-trained LLMs like BERT and GPT [52], while these embeddings are learned independently from decision tasks and may fail to capture domain-related information. Early methods develop rule-based [53] or task-specific [54] intermediate representations to capture task-related semantics, which can require a laborious design and lack scalability. Under an imitation learning paradigm, BC-Z [25] trains a vision-based robotic manipulation policy conditioned on pre-trained language embeddings, and BAKU [27] produces a simple transformer architecture that fuses multi-modal vision-language information and temporal context for multi-task decision-making. To improve language grounding, [55] translates natural language to task language with a referential game, and [26] uses supervised fine-tuning to enable bidirectional translation between language-described skills and rollout data. Overall, the challenges include the inability to capture environment dynamics, knowledge misalignment due to semantic gaps, and limited model scalability. In the paper, we tackle these challenges to fully harness LLMs for building generalist decision agents.

## 3   Method

In this section, we present Text-to-Decision Agent (T2DA), a simple and scalable framework to train generalist policies. Figure 2 illustrates the overall pipeline, followed by detailed implementations in the subsequent subsections. Corresponding algorithm pseudocodes are given in Appendix A.

### 3.1   Problem Statement

We consider a language-conditioned offline meta-RL setting. Tasks follow a distribution $M_k = \langle \mathcal{S}, \mathcal{A}, \mathcal{T}_k, \mathcal{R}_k, \gamma \rangle \sim P(M)$, sharing the same state-action spaces while varying in the reward and state transition functions, i.e., environment dynamics. For each training task $k$, our system receives a user-provided natural language supervision $l_k$ that describes the task (e.g., open the door), paired with an offline dataset $\mathcal{D}_k = \sum_i (s_i^k, a_i^k, r_i^k, s_i^{'k})$ collected by arbitrary behavior policies. The agent can only access the offline datasets $\sum_k (l_k, \mathcal{D}_k)$ to train a generalist policy $\pi(a|s, l)$ to follow language instructions. At test time, natural language is used to reference learned decision perceptions or describe new ones. The agent can perform text-to-decision generation in test environments in a zero-shot manner, given any language instruction $l_{\text{new}}$ at hand. The primary objective is to obtain a highly generalizable policy that achieves strong zero-shot performance on unseen tasks.

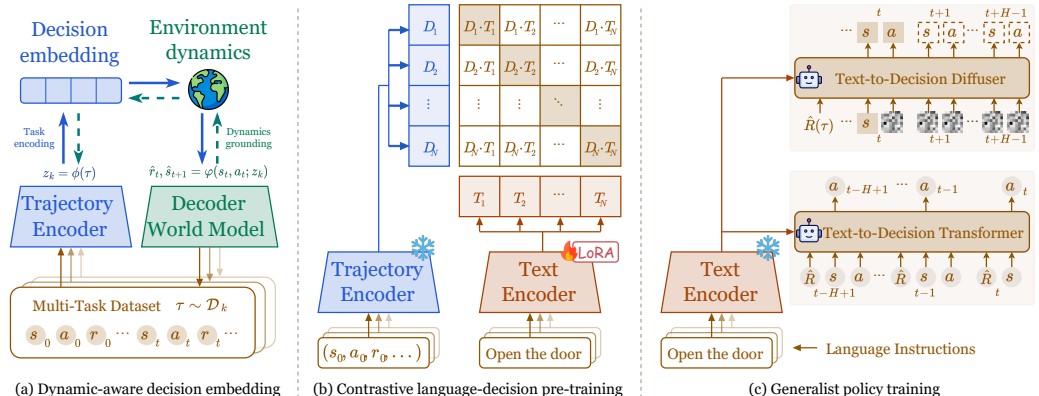

| | | |
|---|---|---|
| (a) Dynamic-aware decision embedding | (b) Contrastive language-decision pre-training | (c) Generalist policy training |

Figure 2: The overall pipeline of T2DA. (a) We encode the multi-task trajectories into dynamics-aware decision embeddings and decode the generalized world model conditioned on that embedding, effectively capturing the environment dynamics. (b) We bridge the semantic gap between decision and text by fine-tuning the text encoder (initialized from popular language models such as CLIP or T5) to align the produced text embeddings with dynamics-aware decision embeddings using contrastive loss. It distills the world model structure from decision embeddings to the text modality, aligning text embeddings to comprehend the environment dynamics. (c) We condition the generalist policy on the aligned text embeddings, and develop scalable implementations with the potential to train decision models at scale: Text-to-Decision Diffuser and Text-to-Decision Transformer. During evaluation, the agent can directly realize zero-shot text-to-decision generation according to textual instructions at hand, enabling high-capacity zero-shot generalization to downstream tasks.

## 3.2 Dynamics-Aware Decision Embedding

Natural language is a complex and unbounded representation of human instruction, and training policies to harness language supervision is nontrivial. A standard approach is to convert language instructions to simpler latent embeddings using popular pre-trained language models that can draw on a wealth of knowledge learned from copious amounts of text. However, directly applying these models to decision domains could fail to solve user-specific problems, as they are typically learned independently of decision-making tasks. A major challenge is that LLMs trained on text corpora typically lack grounding in the interacting environment and fail to capture its dynamics.

For effective grounding, the agent should comprehend environment dynamics of underlying decision tasks. RL typically learns from active interactions with the outer environment. The world model, i.e., the reward and state transition functions $p(s', r|s, a)$, fully characterizes the environment and presents a promising alternative to capturing environment dynamics. Hence, we introduce a generalized world model to encode the multi-task decision data into a dynamics-aware embedding space. We separate the reasoning about the world model into two parts: i) encoding the dynamics-specific information into a latent embedding, and ii) decoding environment dynamics conditioned on that embedding.

First, we use a trajectory encoder $\phi$ to abstract multi-task decision trajectories into a compact embedding $z$ that captures task-specific dynamics. Specifically, we "tokenize" different elements (i.e., state, action, and reward) in the raw sequence $\tau = (s_0, a_0, r_0, ..., s_L, a_L, r_L) \sim \mathcal{D}_k$, by lifting them to a common representation space using element-specific tokenizers $f_\phi^s, f_\phi^a, f_\phi^r$ as

$$\tau_e = (e_0^s,\ e_0^a,\ e_0^r,\ ...,\ e_L^s,\ e_L^a,\ e_L^r), \quad \text{where } e_t^s = f_\phi^s(s_t), e_t^a = f_\phi^a(a_t), e_t^r = f_\phi^r(r_t),\ \forall t \in [0, L]. \quad (1)$$

We use a bi-directional transformer $E_\phi$ to extract the dynamics-aware embedding from input tokens as $z_k = E_\phi(\tau_e)$. The bi-directional structure allows for capturing the forward and inverse dynamics.

Second, we introduce a decoder $\varphi$ containing a reward model $\mathcal{R}_\varphi$ and state transition model $\mathcal{T}_\varphi$. The latent embedding is augmented into the input to predict the instant reward $\hat{r}_t = \mathcal{R}_\varphi(s_t, a_t; z_k)$ and next state $\hat{s}_{t+1} = \mathcal{T}_\varphi(s_t, a_t; z_k)$. The encoder-decoder pipeline is jointly trained by minimizing the reward and state transition prediction error conditioned on the decision embedding as

$$\mathcal{L}(\phi, \varphi) = \mathbb{E}_{\tau \sim \mathcal{D}_k} \left[ \mathbb{E}_{z_k \sim \phi(\tau)} \left[ \mathbb{E}_t \left[ (r_t - \mathcal{R}_\varphi(s_t, a_t; z_k))^2 + (s_{t+1} - \mathcal{T}_\varphi(s_t, a_t; z_k))^2 \right] \right] \right], \forall k. \quad (2)$$

In practice, we randomly sample a trajectory from the dataset as $\tau \sim \mathcal{D}_k$ and use its decision embedding to decode the dynamics of other trajectories in the same dataset as $\tau^* \in \mathcal{D}_k \backslash \tau$. The reason is that, since the embedding has access to the entire trajectory's information, using it to decode the same trajectory can lead to deceptive traps during training. After proper training, we freeze the world model for the subsequent learning phases.

## 3.3 Contrastive Language-Decision Pre-training

To successfully instruct generalist training with human language, we must bridge the semantic gap between text and the "decision modality". Inspired by CLIP [16], we predict which textual description goes with which training task, bridging their semantic gap via contrastive learning. The core idea is to connect the language supervision with corresponding decision embeddings, distilling the world model structure to the text modality and aligning text embeddings to comprehend environment dynamics.

Formally, given a batch of $N$ (trajectory $\tau$, textual task description $l$) pairs, we predict which of the $N \times N$ possible (trajectory, text) pairings across a batch actually occurred. We use the pre-trained trajectory encoder in Sec. 3.2 to extract the dynamics-aware decision embedding from the trajectory as $z = \phi(\tau)$. The text encoder is a transformer-based model $\psi$ that could be initialized from one of a wide variety of popular language models, such as CLIP [16] or T5 [56]. Then, the text embedding $z_T$ is easily derived by tokenizing the textual instruction $l$ and feeding it to the encoder as $z_T = \psi(l)$.

We model the pre-training task as a text-decision multi-modal learning problem by fine-tuning the text encoder while keeping the trajectory encoder fixed. The objective is to maximize the cosine similarity of the decision and text embeddings of the $N$ real pairs in the batch while minimizing the cosine similarity of the embeddings of the $N^2 - N$ incorrect pairings. The two directional similarities between the modalities, which are transposes of each other, are defined as

$$\text{sim}(\tau, l) = \exp(\alpha) \cdot \frac{\phi(\tau)W_D \cdot \psi(l)W_T}{||\phi(\tau)W_D|| \cdot ||\psi(l)W_T||}, \quad \text{sim}(l, \tau) = \exp(\alpha) \cdot \frac{\psi(l)W_T \cdot \phi(\tau)W_D}{||\psi(l)W_T|| \cdot ||\phi(\tau)W_D||}, \quad (3)$$

where $\alpha$ is a learnable temperature controlling the range of similarities, and $W_D$ and $W_T$ are learnable projections that map feature representations of each modality to a joint multi-modal embedding space. The decision-to-text $p(\tau)$ and text-to-decision $p(l)$ similarity scores in the batch are calculated as

$$p(\tau_k) = \frac{e^{\text{sim}(\tau_k, l_k)}}{\sum_{i=1}^N e^{\text{sim}(\tau_k, l_i)}}, \quad p(l_k) = \frac{e^{\text{sim}(l_k, \tau_k)}}{\sum_{i=1}^N e^{\text{sim}(l_k, \tau_i)}}. \quad (4)$$

Let $q(\tau)$ and $q(l)$ denote the ground-truth similarity scores, where negative pairs have a probability of 0 and positive pairs have 1. We optimize a symmetric cross-entropy loss over similarity scores as

$$\mathcal{L}(\psi) = 0.5 \cdot \mathbb{E}_{\tau, l} \left[ \text{CE}\left(p(\tau), q(\tau)\right) + \text{CE}\left(p(l), q(l)\right) \right], \quad (5)$$

where CE is the cross-entropy between two distributions. To effectively adapt the text encoder to the semantics of decision modality at a lightweight computing cost, we employ low-rank adaptation (LoRA) [57] to fine-tune the text encoder with the above contrastive loss. Experiments in Appendix E present the analysis of its efficiency and superiority compared to full-parameter fine-tuning.

This alignment process distills the environment dynamics from the decision embedding to the text modality. The aligned text embeddings not only serve as mere linguistic representations but also establish stronger comprehension of underlying decision tasks, effectively bridging the semantic gap between language supervision and decision dynamics.

## 3.4 Generalist Policy Learning

A widely adopted manner to tackle generalization in RL domains is to condition the policy on some representation that can capture task-relevant information, such as context-based meta-learning [35, 36] and prompt-based methods [12, 14]. Under this unified framework, we formulate the generalist agent as a task-conditioned policy model. There exists a true variable that represents the task identity, and we use a latent representation $h$ to approximate that variable. Then, a base policy $\pi(\cdot, h)$ can be shared across tasks by conditioning on the task representation as $\pi_k(a|s) = \pi(a|s; h_k), \forall k$. Previous studies usually approximate the task representation by acquiring expert decision data [12] or exploring the unseen task first [13]. This restricted form of supervision limits their universality and practicability since expensive high-quality samples or explorations are needed to specify any new task.

Motivated by the recent success of language models, we study a promising alternative that leverages a much broader source of supervision. Natural language is a flexible representation for transferring a variety of human ideas and intentions. Learning from its supervision can not only enhance the representation power but also enable efficient knowledge transfer. We map the textual task description $l$ to a latent embedding $\psi(l)$ via the fine-tuned text encoder in Sec. 3.3, and use it to approximate the task representation as $h \approx \psi(l)$. Then, the task-agnostic generalist policy is approximated by

$$\pi(a \mid s;\ h_k) \approx \pi\left(a \mid s;\ \psi(l_k)\right), \quad \forall k. \tag{6}$$

At test time, the agent can directly perform text-to-decision policy generation according to any textual instruction $l_{\text{new}}$ at hand, enabling high-capacity zero-shot generalization to downstream tasks without the need for expensive decision demonstrations or warmup explorations.

### 3.5   Scalable Implementations

We develop scalable implementations of T2DA using two mainstream conditional generation architectures that hold the promise to train RL models at scale: the decision diffuser [58] and the decision transformer [59], yielding the following T2DA variants.

**Text-to-Decision Diffuser (T2DA-D)**. Inspired by the recent success of text-to-image diffusion models [60, 61], we implement the text-to-decision counterpart by modeling the policy as a return-conditioned diffusion model with action planning. For model simplicity and scalability, we choose to diffuse over the $H$-step state-action trajectory as

$$x_c(\tau) = \begin{bmatrix} s_t & s_{t+1} & \dots & s_{t+H-1} \\ a_t & a_{t+1} & \dots & a_{t+H-1} \end{bmatrix}_c, \tag{7}$$

where $c$ denotes the timestep in the forward diffusion process, and $c=0$ corresponds to the unperturbed data. [1] To enable a generalist policy with natural language supervision, we include the fine-tuned text embedding $\psi(l)$ as an additional condition to the diffusion model. To this end, the generalist policy training is formulated as the standard problem of conditional generative modeling as

$$\max_{\theta} \mathbb{E}_{\tau \sim \mathcal{D}}\left[\log p_{\theta}\left(x_0(\tau) \mid \hat{R}(\tau);\ \psi(l)\right)\right], \tag{8}$$

where $\theta$ denotes diffusion parameters and $\hat{R}$ is the return-to-go. During evaluation, we can cast planning in RL as sampling from the diffuser. At each timestep, we observe a state $s_t$ in the environment, sample an $H$-step trajectory $x_0(\tau)$ with the diffusion process conditioned on a target return-to-go and task prompt $\psi(l)$, execute the first action $a_t$ in $x_0(\tau)$, and transition to the next state $s_{t+1}$. More details can be referred to as in Algorithms 3 and 5 in Appendix A.

**Text-to-Decision Transformer (T2DA-T)**. We also implement T2DA with the causal transformer architecture, drawing upon the simplicity, scalability, and associated advances in language modeling such as BERT and GPT. We prepend a task prompt to the input of the causal transformer to realize generalization across tasks, akin to Prompt-DT [12]. For each task, we take the prompt-augmented trajectory $\tau^+$ as the input, which contains both the fine-tuned text embeddings $\psi(l)$ and the most recent $H$-step history sampled from offline datasets as

$$\tau^+ = \left(\psi(l);\ \hat{R}_{t-H+1}, s_{t-H+1}, a_{t-H+1}, ..., \hat{R}_t, s_t, a_t\right), \tag{9}$$

where the return-to-go $\hat{R}_t$ is the cumulative rewards from the current time step till the end of the episode. Then, the model predicts actions autoregressively using a causal self-attention mask. This architecture introduces only one additional token compared to the standard decision transformer, allowing minimal architecture change and lightweight cost for generalist training. More details can be referred to as in Algorithms 4 and 6 in Appendix A.

---

[1]Decision diffuser diffuses over only states and trains another inverse dynamics model to generate actions. In contrast, we only keep a unified diffusion model to i) ensure simplicity with a single end-to-end model in a multi-task setting, and ii) unlock the potential of diffusion models at scale, rather than depending on additional MLPs to generate part of the information, e.g., actions.

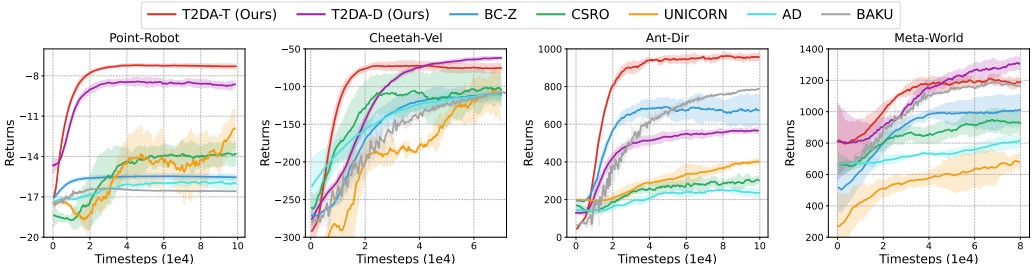

Figure 3: Zero-shot test return curves of T2DA against baselines using Mixed datasets.

Table 1: Zero-shot test returns of T2DA against baselines using Mixed datasets, i.e., numerical results of converged performance from Figure 3. Best results in **bold** and second best underlined.

| Environment | **T2DA-T** | **T2DA-D** | BC-Z | CSRO | UNICORN | AD | BAKU |
|---|---|---|---|---|---|---|---|
| Point-Robot | $-\mathbf{7.2}_{\pm 0.1}$ | $\underline{-8.4}_{\pm 0.2}$ | $-15.5_{\pm 0.2}$ | $-13.7_{\pm 0.5}$ | $-11.4_{\pm 1.3}$ | $-15.8_{\pm 0.2}$ | $-16.4_{\pm 0.1}$ |
| Cheetah-Vel | $\underline{-70.3}_{\pm 3.8}$ | $-\mathbf{60.4}_{\pm 1.8}$ | $-107.4_{\pm 8.5}$ | $-92.6_{\pm 5.9}$ | $-94.1_{\pm 20.0}$ | $-107.6_{\pm 2.3}$ | $-104.5_{\pm 4.8}$ |
| Ant-Dir | $\mathbf{970.3}_{\pm 8.7}$ | $570.6_{\pm 13.1}$ | $700.0_{\pm 43.3}$ | $317.1_{\pm 27.6}$ | $407.3_{\pm 21.6}$ | $268.9_{\pm 3.4}$ | $\underline{786.9}_{\pm 17.3}$ |
| Meta-World | $\underline{1274.5}_{\pm 48.4}$ | $\mathbf{1376.4}_{\pm 39.6}$ | $1053.8_{\pm 33.8}$ | $1005.2_{\pm 69.6}$ | $754.6_{\pm 62.7}$ | $921.0_{\pm 66.9}$ | $1187.6_{\pm 18.2}$ |

# 4 Experiments

We comprehensively evaluate and analyze our method on popular benchmarking domains across datasets of varying qualities, aiming to answer the following research questions:

- Can T2DA achieve consistent performance gain on zero-shot generalization capacity to unseen tasks? We compare it to various types of strong baselines, including offline meta-RL, in-context RL, and language-conditioned policy learning approaches. (Sec. 4.1)

- What is the contribution of each component to T2DA's performance? We ablate both the T2DA-D and T2DA-T architectures to analyze the respective impact of world model pre-training, contrastive language-decision pre-training, and language supervision. (Sec. 4.2)

- How robust is T2DA across diverse settings? We evaluate T2DA against baselines using offline datasets of varying qualities, and assess T2DA's performance when initializing the text encoder from different LLMs. (Sec. 4.3)

**Environments.** We evaluate T2DA on three benchmarks that are widely adopted to assess generalization capacities of RL algorithms: i) the 2D navigation `Point-Robot`; ii) the multi-task MuJoCo locomotion control, containing `Cheetah-Vel` and `Ant-Dir`; and iii) the `Meta-World` platform for robotic manipulation, where a robotic arm is designed to perform a wide range of manipulation tasks, such as close faucet, lock door, open door, and press button. For each domain, we randomly sample a distribution of tasks captioned with text descriptions, and split them into training and test sets. We employ SAC [62] to independently train a single-task policy on each training task for offline dataset collection. We develop three types of offline datasets: `Mixed`, `Medium`, and `Expert`, following the common practice in offline RL [63]. Appendix B presents more details of environments and datasets.

**Baselines.** We compare T2DA to five competitive baselines that cover three representative paradigms in tackling RL generalization: the context-based offline meta-RL approaches, including 1) `CSRO` [13] and 2) `UNICORN` [37]; the in-context RL method of 3) `AD` [42]; and the language-conditioned policy learning methods including 4) `BC-Z` [25] and 5) `BAKU` [27].

For evaluation, all experiments are conducted using five different random seeds. The mean of received return is plotted as the bold line, with 95% bootstrapped confidence intervals of the mean indicated by the shaded region. Appendix D gives detailed implementations of T2DA. Appendix E gives analysis on fine-tuning of the text encoder.

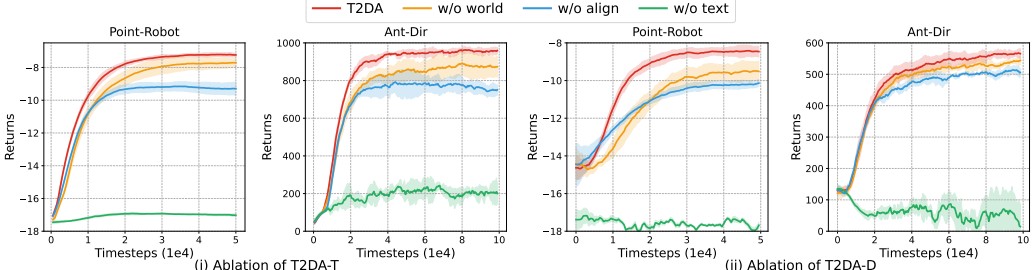

Figure 4: Ablation results using Mixed datasets. `w/o world` omits pre-training the trajectory encoder, `w/o align` omits contrastive pre-training, and `w/o text` omits the language supervision.

Table 2: Numerical results of ablation study on both T2DA-D and T2DA-T using Mixed datasets, i.e., the final test returns after learning convergence from Figure 4. Best results in **bold**.

| Environment | **T2DA-T** | w/o world | w/o align | w/o text | **T2DA-D** | w/o world | w/o align | w/o text |
|---|---|---|---|---|---|---|---|---|
| Point-Robot | $\mathbf{-7.2}_{\pm 0.1}$ | $-7.7_{\pm 0.3}$ | $-9.2_{\pm 0.2}$ | $-16.9_{\pm 0.1}$ | $\mathbf{-8.4}_{\pm 0.3}$ | $-9.5_{\pm 0.4}$ | $-10.1_{\pm 0.2}$ | $-17.2_{\pm 0.4}$ |
| Ant-Dir | $\mathbf{963}.4_{\pm 15.3}$ | $891.0_{\pm 54.4}$ | $792.3_{\pm 48.3}$ | $242.6_{\pm 13.7}$ | $\mathbf{568.0}_{\pm 10.8}$ | $544.1_{\pm 11.1}$ | $513.8_{\pm 11.0}$ | $135.1_{\pm 10.3}$ |

## 4.1 Main Results

We compare our method against various baselines under an aligned zero-shot setting. Figure 3 and Table 1 present test return curves and numerical results of converged performance on various benchmark environments using Mixed datasets. A noteworthy point is that language-conditioned baselines such as BC-Z and BAKU generally achieve better performance than offline meta-RL and in-context RL baselines, especially in harder environments like Ant-Dir and Meta-World. This again validates our motivation of exploring a much broader source of supervision from natural language, harnessing the representation power and knowledge transferability embodied in pre-trained LLMs.

In these diversified environments, both T2DA-D and T2DA-T consistently achieve significantly superior performance regarding learning speed and final asymptotic results compared to the three types of baselines. In most cases, T2DA-D and T2DA-T take the top two rankings for the best and second-best performance, and showcase comparably strong zero-shot generalization capacities on average across all evaluated environments. It highlights the effectiveness of both T2DA implementations using the two mainstream conditional generation architectures. Furthermore, our method typically demonstrates lower variance during generalist policy learning, signifying not only enhanced learning efficiency but also improved training stability.

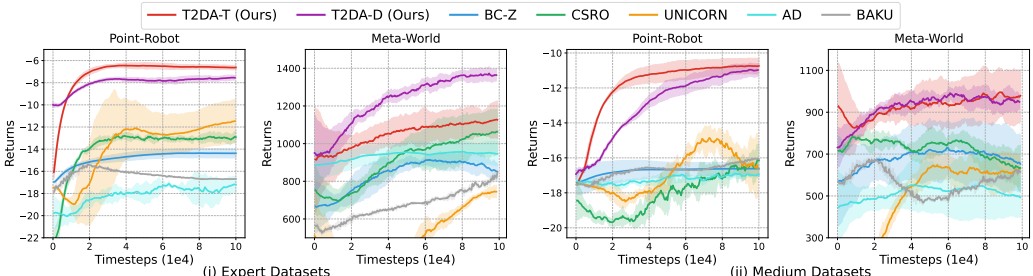

Figure 5: Results of robustness to data quality, where T2DA is compared to baselines using Expert and Medium datasets. T2DA-D and T2DA-T achieve consistent superiority across various datasets.

## 4.2 Ablation Study

We compare T2DA to three ablations: 1) `w/o world`, it eliminates pre-training the trajectory encoder and instead updates it jointly with the text encoder during contrastive pre-training; 2) `w/o align`, it

Table 3: Numerical results of robustness to data quality where T2DA is compared against baselines using Expert and Medium datasets, i.e., the final test returns after learning convergence from Figure 5. Both T2DA-D and T2DA-T achieve consistent superiority across datasets of varying quality. Best results in **bold** and second best underlined.

| **Expert** | T2DA-T | T2DA-D | BC-Z | CSRO | UNICORN | AD | BAKU |
|---|---|---|---|---|---|---|---|
| Point-Robot | $-6.4_{\pm0.2}$ | $-7.5_{\pm0.2}$ | $-14.4_{\pm0.4}$ | $-12.8_{\pm0.2}$ | $-11.5_{\pm1.7}$ | $-17.1_{\pm0.2}$ | $-15.4_{\pm0.1}$ |
| Meta-World | $1142.8_{\pm106.6}$ | $\mathbf{1387.7}_{\pm21.6}$ | $935.5_{\pm26.7}$ | $1087.0_{\pm78.8}$ | $787.1_{\pm16.0}$ | $991.0_{\pm37.7}$ | $833.6_{\pm8.3}$ |

| **Medium** | T2DA-T | T2DA-D | BC-Z | CSRO | UNICORN | AD | BAKU |
|---|---|---|---|---|---|---|---|
| Point-Robot | $\mathbf{-10.7}_{\pm0.4}$ | $-11.0_{\pm0.3}$ | $-16.6_{\pm0.2}$ | $-16.1_{\pm0.8}$ | $-14.9_{\pm0.5}$ | $-16.9_{\pm0.2}$ | $-16.1_{\pm0.1}$ |
| Meta-World | $\mathbf{1033.3}_{\pm130.6}$ | $1025.3_{\pm43.6}$ | $756.9_{\pm103.4}$ | $854.1_{\pm17.9}$ | $715.8_{\pm93.5}$ | $585.7_{\pm126.1}$ | $669.5_{\pm11.4}$ |

omits contrastive language-decision pre-training and fixes the text encoder initialized from pre-trained language models; and 3) `w/o text`, it completely removes the text encoder.

Figure 4 presents ablation results of both architectures. First, the erasure of pre-training the trajectory encoder (T2DA *vs.* w/o world) results in decreased test returns with greater variances. It indicates that capturing environment dynamics via the world model can enable more precise and stable knowledge alignment between text and decision. Second, omitting the language-decision pre-training (w/o world *vs.* w/o align) further decreases the performance. It confirms the existence of a semantic gap between text and decision, whereas the gap can be effectively bridged by contrastive knowledge alignment. Finally, removing the language supervision (w/o align *vs.* w/o text) can produce catastrophic performance with poor zero-shot generalization, highlighting the necessity of leveraging a wealth of knowledge from natural language. In summary, the performance of T2DA declines when any component is omitted, verifying that all components are essential to its final capabilities.

## 4.3 Robustness Study

A reliable framework with scalability should be robust across diverse settings. We assess T2DA's robustness in two critical aspects: data quality and choice of text encoders.

**Robustness to Data Quality.** We compare T2DA against baselines using Expert and Medium offline datasets. As shown in Figure 5 and Table 3, both T2DA-D and T2DA-T consistently outperform baselines on these datasets. Notably, while most baselines suffer significant performance degradation when trained on Medium datasets, T2DA maintains high-capacity generalization despite the lower data quality. It showcases the promising applicability of T2DA in real-world scenarios where the agent often needs to learn from sub-optimal data. In summary, T2DA achieves consistent superiority across datasets of varying quality, yielding excellent robustness.

**Robustness to Text Encoders.** We also investigate T2DA's sensitivity to different representations of language supervision by initializing the text encoder from three distinct language models: CLIP [16], BERT [64], and T5 [56]. As shown in Figure 6, the near-identical performance across all language models highlights that T2DA's effectiveness is independent of the specific text encoder architecture. It suggests that T2DA can effectively leverage the semantic understanding capabilities inherent to different language models while maintaining consistent performance.

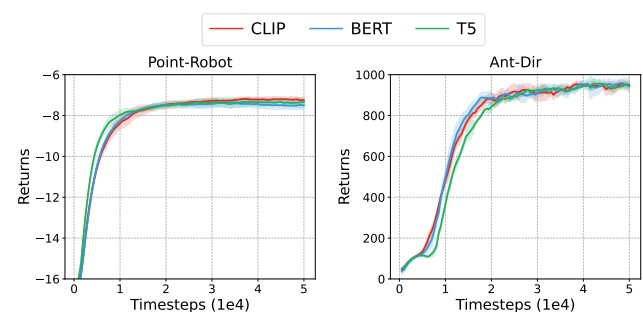

Figure 6: Results of robustness to text encoders where T2DA initializes the text encoder from CLIP, BERT, and T5, showing that T2DA can efficiently harness language knowledge embedded in different language models.

## 4.4 Visualization Insights

We gain deep insights into the knowledge alignment between text and decision through t-SNE visualization on the example Cheetah-Vel task as shown in Figure 7.

**Decision Embeddings**. Raw decision embeddings are derived from a randomly initialized trajectory encoder. In the dynamics-aware embedding space, sample points from different tasks are more clearly distinguished, and similar tasks are grouped more closely. The initially entangled trajectories are transformed into well-separated clusters, highlighting the successful capture of task-specific environment dynamics via the generalized world model.

**Text Embeddings**. Raw text embeddings are derived from pre-trained language models, and are scattered with a vague distribution structure. In the aligned space, text embeddings follow a more distinguished distribution that matches the metric of task similarity, highlighting successful bridging of the semantic gap. Notably, the aligned embeddings form a clear rectilinear distribution from red (low velocity) to blue (high velocity), exactly matching the rectilinear spectrum of target velocities *in a physical sense*. It verifies efficient comprehension of the environment dynamics.

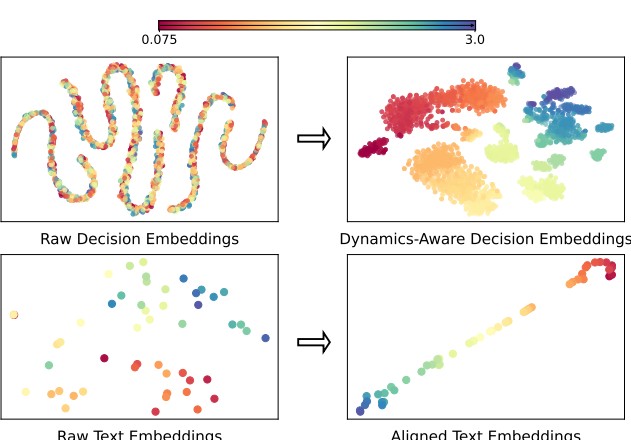

Figure 7: t-SNE visualization of Cheetah-Vel where tasks with target velocities in $[0.075, 3.0]$ are mapped into rainbow-colored points. Top: the evolution from Raw Decision Embeddings to Dynamics-Aware Decision Embeddings, where the initially entangled trajectories are transformed into well-separated clusters that highlight the successful capture of task-specific environment dynamics. Bottom: using dynamics-aware decision embeddings to align text embeddings with contrastive loss. This inspiring finding verifies the effective alignment of text embeddings to comprehend environment dynamics and interpretable language grounding.

## 5 Conclusions, Limitations, and Future Work

In the paper, we tackle offline meta-RL challenge via leveraging a much broader source of supervision from natural language. Improvements in zero-shot generalization capacities highlight the potential impact of our scalable implementations: text-to-decision diffuser and text-to-decision transformer. These findings open up new avenues for building more flexible and accessible generalist agents that can understand and act upon natural language instructions without extensive domain-specific training.

Though, our agent is trained on lightweight datasets compared to popular large models. An essential step is to implement on vast datasets with diversified domains, unleashing the scaling law with the diffusion or transformer architectures. Another step is to deploy the idea of knowledge alignment in VLA models, or deploy the text-to-decision paradigm on real robots toward embodied intelligence.

## Acknowledgements

This work was supported in part by the National Natural Science Foundation of China (Nos. 62376122 and 72394363), and in part by the AI & AI for Science Project of Nanjing University.

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

# Appendix

# A Algorithm Pseudocodes

Based on the implementations in Sec. 3, this appendix gives the brief procedures of each component in T2DA. First, Algorithm 1 presents the pre-training of the generalized world model. Based on the dynamics-aware decision embedding encoded by Algorithm 1, Algorithm 2 shows the contrastive language-decision pre-training that bridges the semantic gap between text and decision, and aligns text embeddings to comprehend the environment dynamics of decision tasks.

Then, Algorithm 3 and Algorithm 4 present the model training processes of Text-to-Decision Diffuser and Text-to-Decision Transformer, respectively, where the generalist policy is conditioned on the text embeddings fine-tuned in Algorithm 2. Finally, Algorithm 5 and Algorithm 6 show zero-shot evaluations on test tasks, where the agent can directly realize text-to-decision generation according to textual instructions at hand.

---

**Algorithm 1:** Pre-training the generalized world model

---

**Input:** Offline datasets $\mathcal{D}_{\text{train}}$;      Batch size $m$

          Trajectory encoder $\phi$;      Reward decoder $\mathcal{R}_\varphi$ and Transition decoder $\mathcal{T}_\varphi$

1 **for** *each iteration* **do**

2    **for** $i = 1, ..., m$ **do**

3       Sample a task $M_k$ and obtain the corresponding dataset $\mathcal{D}_k$ from $\mathcal{D}_{\text{train}}$

4       Sample two trajectories from $\mathcal{D}_k$ as $\tau = (s_0, a_0, r_0, s_1, a_1, r_1, ...)$ and

        $\tau^* = (s_0^*, a_0^*, r_0^*, s_1^*, a_1^*, r_1^*, ...)$

5       Obtain the decision embedding by feeding $\tau$ to the trajectory encoder as $z = \phi(\tau)$

6       **for** $t = 0, 1, ...$ **do**

7         Compute the predicted reward $\hat{r}_t^* = \mathcal{R}_\varphi(s_t^*, a_t^*; z)$ and next state

         $\hat{s}_{t+1}^* = \mathcal{T}_\varphi(s_t^*, a_t^*; z)$ on trajectory $\tau^*$

8       **end**

9    **end**

10   Update $\phi$ and $\varphi$ jointly using loss

     $\mathcal{L}(\phi, \varphi) = \frac{1}{m} \sum_i \sum_t \left[ \left( r_t^* - \mathcal{R}_\phi(s_t^*, a_t^*; z) \right)^2 + \left( s_{t+1}^* - \mathcal{T}_\varphi(s_t^*, a_t^*; z) \right)^2 \right]$

11 **end**

---

**Algorithm 2:** Contrastive Language-Decision Pre-training

---

**Input:** Offline datasets $\mathcal{D}_{\text{train}}$;          Batch size $N$

          Pre-trained trajectory encoder $\phi$;     Text encoder $\psi$

1 Freeze the trajectory encoder $\phi$ pre-trained in Algorithm 1

2 Initialize text encoder $\psi$ from popular pre-trained language models like CLIP or T5

3 **for** *each iteration* **do**

4    Sample a batch of $N$ (trajectory $\tau$, textual task description $l$) pairs from $\mathcal{D}_{\text{train}}$

5    Compute the symmetric similarities between the two modalities, $\text{sim}(\tau, l)$ and $\text{sim}(l, \tau)$, as in Eq. (3)

6    Derive the decision-to-text and text-to-decision similarity scores, $p(\tau)$ and $p(l)$, as in Eq. (4)

7    Update the text encoder $\psi$ using LoRA by minimizing the symmetric cross-entropy loss $\mathcal{L}(\psi)$ as in Eq. (5)

8 **end**

---

# B Evaluation Environments and Dataset Construction

## B.1 The Details of Environments

We evaluate T2DA and all the baselines on three classical benchmarks that are widely adopted to assess the generalization capacity of RL algorithms, containing the following four environments:

- **Point-Robot** is a 2D navigation environment where an agent starts from a fixed origin and navigates to a target location $g$. The agent receives its coordinate position as the observation

---

**Algorithm 3:** Model Training of Text-to-Decision Diffuser

---

**Input:** Offline datasets $\mathcal{D}_{\text{train}} = \sum_k (\mathcal{D}_k, l_k)$;     Pre-trained text encoder $\psi$
        Noise model $\epsilon_\theta$;                     Batch size $N$

1 Freeze the text encoder $\psi$ pre-trained in Algorithm 2
2 **while** *not converged* **do**
3 |     Sample a batch of $N$ $\left(\text{trajectory } x_0(\tau^k), \text{language supervision } l_k\right)$ pairs from different tasks in $\mathcal{D}_{\text{train}}$
4 |     $c \sim \text{Uniform}(\{1, ..., C\})$
5 |     $\epsilon \sim \mathcal{N}(0, I)$
6 |     $x_c(\tau_{s_t}^k) \leftarrow s_t^k$         // Constrain the first state of the plan
7 |     Take gradient descent step on $\nabla_\theta \sum_{k=1}^N \left\| \epsilon - \epsilon_\theta \left( x_c(\tau^k), \hat{R}(\tau), \psi(l_k), c \right) \right\|^2$
8 **end**

---

---

**Algorithm 4:** Model Training of Text-to-Decision Transformer

---

**Input:** Offline datasets $\mathcal{D}_{\text{train}} = \sum_k (\mathcal{D}_k, l_k)$;     Pre-trained text encoder $\psi$
        Causal transformer $F_\theta$;                 Batch size $N$

1 Freeze the text encoder $\psi$ pre-trained in Algorithm 2
2 **while** *not converged* **do**
3 |     Sample a batch of $N$ (trajectory $\tau_k$, language supervision $l_k$) pairs from different tasks in $\mathcal{D}_{\text{train}}$
4 |     Get a batch of inputs $\mathcal{B} = \{\tau_k^+\}_{k=1}^N$, where
        $$\tau_k^+ = \left( \psi(l_k), \hat{R}_{t-H+1}^k, s_{t-H+1}^k, a_{t-H+1}^k, ..., \hat{R}_t^k, s_t^k, a_t^k \right)$$
5 |     $a^{\text{pred}} = F_\theta(\tau_k^+), \quad \forall \tau_k^+ \in \mathcal{B}$
6 |     Minimize loss $\mathcal{L}(\theta) = \frac{1}{N} \sum_{\tau_k^+ \in \mathcal{B}} \left( a - a^{\text{pred}} \right)^2$
7 **end**

---

---

**Algorithm 5:** Zero-Shot Evaluation of Text-to-Decision Diffuser

---

**Input:** Language instruction $l_{\text{new}}$;         Pre-trained text encoder $\psi$
        Trained noise model $\epsilon_\theta$;         Diffusion steps $C$;                 Guidance scale $w$

1 Obtain the text embedding from language instruction as $\psi(l_{\text{new}})$
2 **while** *not done* **do**
3 |     Observe state $s_t$; Initialize $x_C(\tau) \sim \mathcal{N}(0, \alpha I)$
4 |     **for** $c = C, ..., 1$ **do**
5 | |         $x_c(\tau_{s_t}) \leftarrow s_t$         // Constrain the first state of the plan
6 | |         $\hat{\epsilon} \leftarrow \epsilon_\theta \left( x_c(\tau), c \right) + w \left[ \epsilon_\theta \left( x_c(\tau), \hat{R}(\tau), \psi(l_{\text{new}}), c \right) - \epsilon_\theta \left( x_c(\tau), c \right) \right]$         // Classifier-free guidance
7 | |         $(\mu_{c-1}, \Sigma_{c-1}) \leftarrow \text{Denoise}(x_c(\tau), \hat{\epsilon})$
8 | |         $x_{c-1}(\tau) \sim \mathcal{N}(\mu_{c-1}, \alpha \Sigma_{c-1})$
9 |     **end**
10 |     Execute first action of plan as $a_t \leftarrow x_0(\tau_{a_t})$
11 |     $t \leftarrow t + 1$
12 **end**

---

and outputs actions in the range of $[-0.1, 0.1]^2$, representing displacement in the $X$- and $Y$-axis directions. The reward is the negative Euclidean distance to the goal: $r_t = -||s_t - g||_2$, where $s_t$ denotes the current position. The maximal episode step is set to 20. Tasks differ in the target location $g$ that is randomly sampled within a unit square of $[-0.5, 0.5]^2$. Each task is captioned by a text description as "Please navigate to the goal position of $g$".

- **Cheetah-Vel** is a multi-task MuJoCo environment where a planar cheetah robot aims to run a target velocity $v_g$ along the $X$-axis. The reward function combines a quadratic control cost and a

---

**Algorithm 6:** Zero-Shot Evaluation of Text-to-Decision Transformer

---

**Input:** Language instruction $l_{\text{new}}$;  Pre-trained text encoder $\psi$
  Trained casual transformer $F_\theta$;  Target return $G^*$

1 Obtain the text embedding from language instruction as $\psi(l_{\text{new}})$

2 Initialize desired return $\hat{R} = G^*$

3 **for** *each timestep $t$* **do**

4 |  Observe state $s_t$

5 |  Obtain the $H$-step history trajectory as $\tau = \left(\hat{R}_{t-H+1}, s_{t-H+1}, a_{t-H+1}, ..., \hat{R}_t, s_t\right)$

6 |  Augment the trajectory with the text embedding as $\tau^+ = (\psi(l_{\text{new}}), \tau)$

7 |  Get action $a_t = F_\theta(\tau^+)$

8 |  Step env. and get feedback $s, a, r, \hat{R} \leftarrow \hat{R} - r$

9 |  Append $(\hat{R}, s, a)$ to recent history $\tau$

10 **end**

---

velocity matching term as $r_t = -0.05||a_t||_2 - |v_t - v_g|$, where $a_t$ and $v_t$ represent the action and current velocity, respectively. The maximal episode step is set to 200. Tasks differ in the target velocity $v_g$ that is randomly sampled from a uniform distribution as $v_g \sim \text{Uniform}[0.075, 3.0]$. Each task is captioned by a text description as "Please run at the target velocity of $v_g$".

- **Ant-Dir** is also a multi-task MuJoCo environment where a 3D ant robot navigates toward a target direction $\theta$. The primary reward component measures directional alignmentas $r_{\text{forward}} = v_x \cos \theta + v_y \sin \theta$, where $(v_x, v_y)$ represents the horizontal and vertical velocities. The final reward incorporates additional terms for the control cost, contact penalty, and survival bonus. The maximal episode step is set to 200. Tasks differ in the goal direction $\theta$ that is uniformly sampled from the full space as $\theta \sim \text{Uniform}[0, 2\pi]$. Each task is captioned by a text description as "Please walk toward the target direction of $\theta$".

- **Meta-World** comprises 50 robotic manipulation tasks where a robotic arm interacts with various objects on a tabletop. Each task's reward function (ranging from 0 to 10) combines multiple components for fundamental behaviors (such as reaching, grasping, and placing), with 10 indicating successful completion. The maximal episode step is set to 200. Table 4 illustrates the detailed description of the training and test tasks. More details can be found in [65].

For each domain of Point-Robot, Cheetah-Vel, and Ant-Dir, we sample 50 tasks in total and split them into 45 training tasks and 5 test tasks. For Meta-World, we use 18 training tasks and 4 test tasks as detailed as shown in Table 4.

### B.2   The Details of Datasets

We employ the soft actor-critic (SAC) algorithm [62] to independently train a policy for each task and save policy checkpoints at different training steps. The hyperparameters used for SAC training across all environments are listed in Table 5. We consider three types of datasets as

- **Medium**: We load the checkpoint of a medium policy that achieves approximately half the performance of expert policies, and use the medium policy to generate a number of trajectories to construct the Medium dataset.

- **Expert**: We load the checkpoint of an expert policy that achieves the highest return after the convergence of training, and use the expert policy to collect several trajectories to construct the Expert dataset.

- **Mixed**: We load all saved checkpoints of policies at different training steps, and use these policies to generate a diverse set of trajectories to construct the Mixed dataset.

Specifically, we collect 200 trajectories for each kind of dataset in all evaluated environments.

Table 4: Details of the training and test tasks of Meta-World.

| Training Task | Description |
|---|---|
| faucet-close | rotate faucet clockwise |
| door-lock | lock door by rotating clockwise |
| door-unlock | unlock door by rotating counter-clockwise |
| window-close | push and close window |
| window-open | push and open window |
| coffee-button | push button on coffee machine |
| drawer-open | open drawer |
| door-open | open door with revolving joint |
| button-press | press button |
| button-press-topdown | press button from top |
| button-press-topdown-wall | bypass wall and press button from top |
| button-press-wall | bypass wall and press button |
| handle-press | press handle down |
| handle-pull | pull handle up |
| plate-slide-back | slide plate back |
| plate-slide-side | slide plate side |
| plate-slide | slide plate |
| plate-slide-back-side | slide plate back side |

| Test Task | Description |
|---|---|
| faucet-open | rotate faucet counter-clockwise |
| drawer-close | push and close drawer |
| reach-wall | bypass wall and reach goal |
| handle-press-side | press handle down sideways |
| handle-pull-side | Pull a handle up sideways |
| hand-insert | Insert the gripper into a hole |

Table 5: Hyperparameters of SAC used to collect multi-task offline datasets.

| Environments | Training steps | Warmup steps | Save frequency | Learning rate | Soft update | Discount factor | Entropy ratio |
|---|---|---|---|---|---|---|---|
| Point-Robot | 2000 | 100 | 40 | 3e-4 | 0.005 | 0.99 | 0.2 |
| Cheetah-Vel | 500000 | 2000 | 10000 | 3e-4 | 0.005 | 0.99 | 0.2 |
| Ant-Dir | 500000 | 2000 | 10000 | 3e-4 | 0.005 | 0.99 | 0.2 |
| Meta-World | 1000000 | 5000 | 100 | 1e-3 | 0.005 | 0.99 | 0.2 |

## C  Baseline Methods

This section gives details of the competitive baselines, including context-based offline meta-RL (OMRL), in-context RL, and language-conditioned policy learning approaches. These baselines are thoughtfully selected to cover the main studies that tackle the generalization problem of RL algorithms. The baselines are introduced as follows:

- **CSRO** [13], **C**ontext **S**hift **R**eduction for **O**MRL. It addresses the context shift problem in offline meta-RL through two key components: 1) During meta-training, it employs a max-min mutual information representation learning mechanism to minimize mutual information between the task representation and the behavior policy while maximizing mutual information between the task representation and the task information; 2) During meta-test, it introduces a non-prior context collection strategy to first randomly explore the environment and then gradually update the task representation.

- **UNICORN** [37], **Uni**fied Information Theoretic Framework of **C**ontext-Based **O**ffline Meta-**R**einforcement Lear**n**ing. It provides a general framework to unify several context-based offline meta-RL algorithms by proving that they optimize different bounds of mutual information between

the task variable $M$ and latent representation $Z$. Based on this theoretical insight, it proposes a supervised and a self-supervised implementation of the mutual information derivation $I(Z; M)$.

- **AD** [42], **A**lgorithm **D**istillation. It distills RL algorithms into neural networks by modeling their training histories with a causal sequence model. Using a dataset of learning histories generated by a source RL algorithm, AD trains a causal transformer to autoregressively predict actions given their preceding learning histories as the context, enabling in-context policy improvement without updating network parameters.

- **BC-Z** [25], **B**ehavior **C**loning **Z**. It is an imitation learning system designed for zero-shot generalization to novel vision-based manipulation tasks. It flexibly conditions the policy on pre-trained embeddings of natural language or video of human-performing tasks. To align BC-Z to the benchmarks investigated in this paper, we replace the video data with RL datasets $\mathcal{D} = \sum_i(s_i, a_i, r_i, s_i')$ and modify its video encoder to a trajectory encoder $\phi$ identical to that used in T2DA. This adjustment for a fair comparison results in the optimization objective: $\min \sum_{\text{task } k} \sum_{(s,a) \sim \mathcal{D}} [-\log \pi(a|s, \psi(l_k)) + D_{\cos}(\psi(l_k), \phi(\tau_k))]$, where $D_{\cos}$ denotes the cosine distance.

- **BAKU** [27], an efficient transformer architecture for multi-task policy learning that improves upon prior work in offline imitation learning. It carefully integrates multiple key components: observation trunks for multi-modal fusion, action chunking for smoother control, multi-sensory observation processing, and modular action heads for flexible prediction. The model is optimized through behavior cloning with the objective: $\min \sum_{\text{task } k} \sum_{(o,a) \sim \mathcal{D}} |a - \pi(a|o, T_k)|^2$, where $o$ represents multi-modal observations and $T_k$ denotes the task instruction. For a fair comparison, we adjust BAKU to the language-conditioned zero-shot setting.

- **Prompt-DT** [12], **Prompt**-based **D**ecision **T**ransformer. It is a DT-based offline meta-RL method that leverages the transformer architecture and prompting mechanisms to enable few-shot adaptation. It introduces trajectory prompts, which consist of short demonstration segments from the target task, to encode task-specific context and guide policy generation. Without requiring any fine-tuning on unseen tasks, Prompt-DT achieves strong few-shot performance through expert demonstrations in contructed prompts.

- **MetaDiffuser** [66], a diffusion-based offline eta-RL method that considers the generalization problem as conditional trajectory generation task with contextual representation. It trains a context conditioned diffusion model to generate task-oriented trajectories for planning. To further enhance the dynamics consistency of the generated trajectories while encouraging trajectories to achieve high returns, it introduces a dual-guided module in the sampling process of the diffusion model.

- **Meta-DT** [14], **Meta D**ecision **T**ransformer. It harnesses the sequential modeling capability of the transformer architecture and enables robust task representation learning through a disentangled world model, thereby achieving efficient generalization in offline meta-RL. It learns a compact task representation via context-aware world model, used as a contextual condition to guide task-oriented sequence generation. Complementary to the task representation, it selects a self-guided trajectory segment from online interactions to further exploit the architectural inductive bias. Thereby, Meta-DT exhibits strong few- and zero-shot performance without expert demonstrations.

For a fair comparison, all baselines are adjusted to an aligned zero-shot setting, where no warm-start data is available before policy evaluation. All these baselines can only use samples generated by the trained meta-policy during evaluation to infer task representations.

## D  Implementation Details of T2DA

### D.1  Network Architecture and Hyperparameters

- **World Model**. We implement the generalized world model using simple architecture: a trajectory encoder and two decoders for the reward and state transition prediction. Specifically, the trajectory encoder consists of a transformer encoder with 6 layers, where each layer contains a multi-head self-attention module (8 heads) and a feed-forward network with ReLU activation. States, actions, and rewards in the raw trajectory are first tokenized into 256-dimensional vectors through separate linear layers. These embeddings are concatenated and processed by the transformer encoder to produce a 256-dimensional decision embedding $z$ via mean pooling and linear projection. Two MLPs serve as decoders: a reward decoder that predicts the instant reward $r$ given the tuple

$(s, a, s'; z)$, and a state transition decoder that predicts the next state $s'$ given the tuple $(s, a, r; z)$. Both MLPs utilize two hidden layers of 256 dimensions each.

- **Text Encoder**. We initialize text encoders from pre-trained CLIP [16], BERT [64], and T5 [56] in this paper. Specifically, we use `openai/clip-vit-base-patch32`, `google-bert/bert-base-uncased`, and `google-t5/t5-small`, respectively.

- **Diffusion Architecture of T2DA**. We implement the diffusion architecture of T2DA-D based on the decision diffuser [58] framework [2]. Following DiT [67], we replace U-Net backbone with a transformer to represent the noise model $\epsilon_\theta$. Each block employs a multi-head self-attention module followed by a feed-forward network with GELU activation [68], using adaptive layer normalization (adaLN) for conditioning. The model processes state-action trajectories through linear embeddings, which are combined with positional embeddings derived from RL timesteps. The diffusion time embeddings are encoded via sinusoidal embeddings processed through an MLP. For conditioning, text embeddings and returns-to-go are concatenated with diffusion time embeddings. The detailed configurations and hyperparameters of T2DA-D are presented in Table 7.

- **Transformer Architecture of T2DA**. We implement the transformer architecture of T2DA-T based on the decision transformer [59] framework [3]. Specifically, we employ modality-specific embeddings for states, actions, returns-to-go, and timesteps. The timestep embeddings serve as positional encodings and are added to each token embedding. These tokens are concatenated with text embedding of task description and fed into a GPT architecture which predicts actions autoregressively using a causal self-attention mask. The detailed configurations and hyperparameters of T2DA-T are presented in Table 6.

Table 6: Configurations and hyperparameters in the training process of T2DA-T.

| Configuration | Point-Robot | Cheetah-Vel | Ant-Dir | Meta-World |
|---|---|---|---|---|
| layers num | 3 | 3 | 3 | 3 |
| attention head num | 1 | 1 | 1 | 1 |
| embedding dim | 128 | 128 | 128 | 128 |
| horizon | 20 | 200 | 40 | 50 |
| training steps | 100000 | 70000 | 100000 | 100000 |
| learning rate | 1e-4 | 2e-4 | 1e-4 | 1e-4 |

Table 7: Configurations and hyperparameters in the training process of T2DA-D.

| Configuration | Point-Robot | Cheetah-Vel | Ant-Dir | Meta-World |
|---|---|---|---|---|
| DiT layers num | 4 | 4 | 8 | 4 |
| DiT attention head num | 8 | 8 | 8 | 8 |
| embedding dim | 128 | 128 | 256 | 128 |
| horizon | 10 | 50 | 40 | 50 |
| diffusion steps | 20 | 20 | 20 | 20 |
| training steps | 100000 | 70000 | 100000 | 100000 |
| learning rate | 5e-5 | 1e-4 | 1e-4 | 1e-4 |

## D.2 Computation

We train our models on one Nvidia RTX4080 GPU with the Intel Core i9-10900X CPU and 256G RAM. Dynamic-aware decision embedding and constrastive language-decision pre-training cost about 0.5 hours. The generalist policy training cost about 0.5-3 hours depending on the complexity of the environment.

---

[2] `https://github.com/anuragajay/decision-diffuser`
[3] `https://github.com/kzl/decision-transformer`

# E    Analysis of Parameter-Efficient Fine-tuning

As stated in Sec. 3.3, we employ LoRA, a classical parameter-efficient fine-tuning method, to fine-tune the text encoder at a lightweight cost. As shown in Table 8, the number of trainable parameters is substantially smaller than full-parameter fine-tuning (less than 1%), enabling much more efficient computation and memory usage. To investigate whether a limited number of trainable parameters constrains the capacity for knowledge alignment, we evaluate T2DA's performance by comparing LoRA fine-tuning with full-parameter fine-tuning during contrastive language-decision pre-training. Figure 8 illustrates the results on representative environments with the text encoder initialized from CLIP. Obviously, the two fine-tuning approaches yield nearly identical performance, highlighting T2DA's superior parameter efficiency of T2DA without sacrificing effectiveness.

Table 8: The number and memory usage of parameters for LoRA fine-tuning and full-parameter fine-tuning.

| Method | Trainable parameters (Memory) | Total parameters (Memory) | Percentage |
|---|---|---|---|
| LoRA Tuning | 589,824 (2.25MB) | 63,755,776 (243.21MB) | 0.93% |
| Full Tuning | 63,165,952 (240.96MB) | 63,165,952 (240.96MB) | 100% |

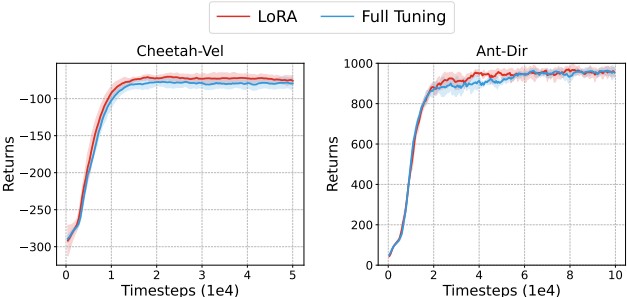

Figure 8: Test return curves of T2DA using LoRA-tuned and fully-tuned CLIP text encoders. With less than 1% parameter count, T2DA using LoRA fine-tuning yields nearly identical performance compared to full-parameter fine-tuning, highlighting T2DA's superior parameter efficiency without sacrificing effectiveness.

# F    Analysis of Training Pipline

T2DA separately trains three key components: (a) `decision embedding`, (b) `knowledge alignment`, and (c) `generalist policy`. To demonstrate the effectiveness of this modular approach, we conducted ablation studies comparing our method against two alternatives: jointly training (a)+(b), and jointly training all three components (a)+(b)+(c) simultaneously. As shown in Table 9, our separate training pipeline achieves substantial performance gains of 20%-56% across different environments. This modular architecture not only provides a more stable training process but also offers superior scalability, as individual components can be easily substituted or upgraded independently without affecting the entire system.

Table 9: Ablation study results on modular architecture. Best results in **bold**.

| Environment | **T2DA** | (a)+(b) | (a)+(b)+(c) |
|---|---|---|---|
| Point-Robot | $-\mathbf{7.2}$ | $-8.9$ | $-16.3$ |
| Meta-World | **1274.5** | 1008 | 817.3 |

## G Meta-RL Baselines with Goal Information

For comprehensive evaluation, we enhanced meta-RL baselines by incorporating goal information into their observation space. This approach provides a standardized framework where all compared methods have access to equivalent task information.

The results in Table 10 demonstrate T2DA's performance advantages over the goal-enhanced baselines, further validating the effectiveness of leveraging natural language supervision. This goal-integrated setting was maintained consistently for all meta-RL baseline comparisons throughout the paper.

Table 10: Performance comparison with goal-enhanced meta-RL baselines. Best results in **bold**.

| Environment | T2DA | CSRO (w/ goal) | UNICORN (w/ goal) |
|---|---|---|---|
| Cheetah-Vel | $-\mathbf{60.4 \pm 1.8}$ | $-84.0 \pm 4.3$ | $-82.7 \pm 9.2$ |
| Ant-Dir | $\mathbf{970.3 \pm 8.7}$ | $384.4 \pm 30.1$ | $427.1 \pm 25.3$ |

## H Performance on Unstructured Language Instructions

This section presents an evaluation of the method's generalization capabilities when confronted with less structured, ambiguous, or noisy natural language instructions. Experiments were conducted where tasks were captioned using two additional types of textual descriptions.

Task descriptions were reconstructed in two distinct linguistic styles:

- **Noisy style:** *"Um, there's this location I think... somewhere around {round(number=task[0], ndigits=2)}-ish in the x direction? And maybe {round(number=task[1], ndigits=2)} or so up?"*

- **Conversational style:** *"Could you move over to the spot that's roughly {round(number=task[0], ndigits=2)} units to the right and {round(number=task[1], ndigits=2)} units up? Thanks!"*

Results demonstrate T2DA's consistent performance across these different textual description formats:

Table 11: Performance comparison across different instruction styles.

| Environment | T2DA | T2DA (Noisy) | T2DA (conversational) |
|---|---|---|---|
| Point-Robot | $-7.2 \pm 0.1$ | $-7.6 \pm 0.1$ | $-8.0 \pm 0.1$ |

These findings highlight the robustness of leveraging natural language supervision, as the language representations effectively capture task-relevant information and successfully extrapolate meta-level knowledge across tasks presented with varying textual descriptions.

## I Experimental Results on Humanoid

This section presents experimental results on stimulated humanoid robot benchmark [69]. Table 12 shows the converged performance of T2DA and baselines using Mixed datasets under aligned zero-shot setting. The results demonstrate that T2DA consistently outperforms baselines in this more challenging environment.

Table 12: Performance comparison of different methods on Humanoid. Best results in **bold** and second best underlined.

| Method | T2DA-T | T2DA-D | BC-Z | DT | DD | CSRO | UNICORN | AD |
|---|---|---|---|---|---|---|---|---|
| Returns | **610.4** | 575.4 | 531.7 | 520.3 | 381.3 | 520.3 | 505.6 | 427.4 |

## J  Comparison to More OMRL Baselines

In this section, we compare T2DA to three additional OMRL baselines: Meta-DT [14], MetaDiffuser [66], and Prompt-DT [12]. More details about these baselines can be found in Appendix C. The results in Table 13 demonstrate T2DA's consistent superiority over these OMRL baselines.

Table 13: Performance comparison with OMRL baselines. Best results in **bold**.

| Environment | **T2DA-T** | **T2DA-D** | Meta-DT | MetaDiffuser | Prompt-DT |
|---|---|---|---|---|---|
| Point-Robot | $-\mathbf{7.2} \pm \mathbf{0.1}$ | $-8.4 \pm 0.2$ | $-8.4 \pm 0.4$ | $-11.7 \pm 0.6$ | $-14.3 \pm 1.7$ |
| Cheetah-Vel | $-70.3 \pm 3.8$ | $-\mathbf{60.4} \pm \mathbf{1.8}$ | $-78.2 \pm 5.3$ | $-86.5 \pm 4.8$ | $-94.6 \pm 8.1$ |
| Ant-Dir | $\mathbf{970.3} \pm \mathbf{8.7}$ | $570.6 \pm 13.1$ | $922.1 \pm 11.9$ | $858.5 \pm 42.5$ | $895.2 \pm 17.3$ |

## K  Limitations

We tackle the offline meta-RL challenge via leveraging the supervision from natural language. Our scalable implementations demonstrate that the generalist RL agent can understand and act upon natural language instructions in a zero-shot manner. While our approach uses relatively lightweight datasets compared to large models, it demonstrates promising results. An essential step of future work is to implement on vast datasets with diversified domains and deploy the text-to-decision paradigm on real robots toward embodied intelligence.

