# OpenReview forum: "Text-to-Decision Agent: Offline Meta-Reinforcement Learning from Natural Language Supervision"
_NeurIPS.cc/2025/Conference — NeurIPS 2025 poster_

### Official Review · Reviewer_EUfm · 2025-06-10

**Clarity:** 3
**Significance:** 2
**Originality:** 1
**Rating:** 4
**Confidence:** 5

**Summary:**

The paper proposes the Text-to-Deicision Agent (T2DA) framework, designed to tackle the offline mtea-RL problem by supervising the policy using natural language. While prior works typically require expert trajectories or environment interactions to infer the task embeddings, the proposed method instead leverages language inputs to directly guide policy inference.

**Questions:**

1. Can authors report the “success rate” for Meta-World environment?

2. Why T2DA outperforms other context-based offline meta-RL approaches (CSRO, UNICORN)? Given that T2DA simply aligns language embeddings with decision embeddings, it is not evident what fundamentally enables its superior performance.

3. Similarly, it is unclear why T2DA maintains superior performance despite the use of lower-quality data, as shown in Figure 5.

**Ethical Concerns:**

["NO or VERY MINOR ethics concerns only"]

**Final Justification:**

Most of my concerns have been addressed through discussion. Accordingly, I have increased my score

**Limitations:**

Yes. Limitations are discussed in Section 5.

**Quality:**

2

**Strengths And Weaknesses:**

1. **Lack of technical challenge**: The main weakness of the paper lies in its problem formulation, which lacks technical depth or significant challenge. The core task is reduced to aligning text embeddings with decision embeddings, which, in its current form, appears to be a relatively straightforward application of existing techniques without addressing more fundamental or difficult aspects of the problem domain.

2. **Lack of novelty**: The individual components of T2DA lack originality. (a) “dynamics-aware decision embedding” resembles standard techniques for task embedding widely adopted in meta-RL literature [1,2,3]. (b) “contrastive language-decision pre-training” simply follows CLIP method without significant methodological innovation.

    [1] VariBAD: A Very Good Method for Bayes-Adaptive Deep RL via Meta-Learning (ICLR20)

    [2] Meta-Reinfrocement Learning Based on Self-Supervised Task Representation Learning (AAAI23)

    [3] MetaDiffuser: Diffusion Model as Conditional Planner for Offline Meta-RL (ICML23)

3. **Design choice for dynamics-aware decision embedding**: Did the authors consider alternative methods for learning task embedings, such as mutual information optimization [4,5] or contrastive representation learning [6] ? It is unclear whether the proposed dynamics-aware approach is the most effective, especially given that recent studies often adopt information-theoretic or contrastive objectives.

    [4] Context Shift Reduction for Offline Meta-Reinfrocement Learning (NeurIPS23)

    [5] Generalizable Task Representation Learning for Offline Meta-Reinforcement Learning with Data Limitation (AAAI24)

    [6] Robust Task Representation for Offline Meta-Reinforcement Learning via Constrastive Learning (ICML22)

4. **Baseline choices**: Some essential language-based policy learning baselines are missing [7,8]. Including such works would provide a more comprehensive evaluation, especially given that original meta-RL baselines used in the paper (CSRO, UNICorn, and AD) do not incorporate language inputs.

    [7] LISA: Learning Interpretable Skill Abstractions from Language (NeurIPS22)

    [8] Language Control Diffusion: Efficiency Scaling Through Space, Time and Tasks (ICLR24)

---

> ### Author Rebuttal · Authors · 2025-07-30
>
> ***
>
> **`Q1. Lack of technical challenge.`**
>
> A1. We disagree with you. Foremost, **we address the key challenge in OMRL** that usually requires either high-quality samples or warmup explorations to infer task beliefs for generalization. To omit these expensive supervision signals, we propose a novel solution using natural language supervision. Then, **we address three fundamental challenges in learning generalist decision agents from natural language supervision**:
>
> - LLMs failing to capture any environment dynamics (we learn dynamics-aware decision embeddings);
> - Knowledge misalignment due to the semantic gap between text and decision (we align text embeddings to comprehend environment dynamics);
> - The necessity of scalable implementations (we develop scalable implementations of text-decision diffuser and text-to-decision transformer).
>
> Through this language-supervised OMRL framework, **T2DA gains a 16% to 200% improvement** over various types of baseline methods. The ablation study also verifies the effectiveness of leveraging language supervision, capturing environment dynamics, and aligning the text-decision semantic gap. In summary, **the significant results validate the effectiveness of tackling the above-mentioned challenges**.
>
> ***
>
> **`Q2. Lack of novelty.`**
>
> A2. We disagree with you. **T2DA is the first to align decision and language embeddings through world model dynamics**. Also, we develop **two new implementations** for realizing language-conditioned generation: text-to-decision diffuser and text-to-decision transformer. Next, we continue to defend our novelty from various aspects as
>
> - **Our dynamics-aware decision embedding extends beyond standard techniques for task embedding**. To our knowledge, the way of using an encoder-decoder world model to learn task embedding was first proposed by VariBAD [1] in meta-RL literature. Existing works [1-3] usually compute a local task embedding $z_t$ per timestep, and use $z_t$ to decode the same trajectory. In contrast, we learn a single-step task embedding $z$ for an entire trajectory, and use $z$ to decode the dynamics of other trajectories. **Our task embedding is more computationally efficient, more generalizable, and more applicable for text-decision alignment**.
> - In literature, it is very common for follow-up studies to adopt standard techniques, while we can still justify the novelty through significant modifications or tackling significant challenges for new problems. For example, MoSS [2] adopts the VariBAD style and improves its task embedding by contrastive learning. MetaDiffuser [3] also adopts the VariBAD style and tackles the diffusion-based OMRL problem. **While our method adopts the basic principle of VariBAD to learn task embedding, we make an innovative improvement upon it and use it to tackle the new challenge for text-to-decision alignment.**
> - **Our novelty of contrastive language-decision pre-training is to extend the CLIP principle to RL domains**, aligning text embeddings to comprehend environment dynamics (**rather than proposing a new CLIP method**). CLIP is a very popular and efficient method for contrastive alignment between different modalities, and has been successfully extended to various domains like VideoCLIP [4], PointCLIP [5], and DiffusionCLIP [6].
>
> In summary, our method is not a straightforward application of existing techniques. **We focus on addressing key challenges in OMRL from language supervision, with significant extensions of existing techniques like VariBAD and CLIP.**
>
> [1] VariBAD: A Very Good Method for Bayes-Adaptive Deep RL via Meta-Learning, ICLR 2020.
>
> [2] Meta-RL Based on Self-Supervised Task Representation Learning, AAAI 2023.
>
> [3] MetaDiffuser: Diffusion Model as Conditional Planner for Offline Meta-RL, ICML 2023.
>
> [4] VideoCLIP: Contrastive Pre-training for Zero-shot Video-Text Understanding, EMNLP 2021.
>
> [5] PointCLIP: Point Cloud Understanding by CLIP, CVPR 2022.
>
> [6] DiffusionCLIP: Text-Guided Diffusion Models for Robust Image Manipulation, CVPR 2022.
>
> ***
>
> **`Q3. Design choice for dynamics-aware decision embedding.`**
>
> A3. Since we aim to align text embeddings to comprehend the environment dynamics, we use the world model to learn task embeddings rather than information-theoretic or contrastive objectives like in CSRO or CORRO. **Through the encoder-decoder world model structure, we can explicitly distill the environment dynamics into the obtained task embedding**. This is the key property that alternative methods do not possess.
>
> In experiments, we have included CSRO and UNICORN [7] as baseines. UNICORN can be considered an improvement over CORRO. Section 4.1 Main Results show that **T2DA gains a 36% to 200% improvement over CSRO and UNICORN**. Further, we have conducted a new ablation experiment where we use the CORRO’s contrastive approach to learn task embeddings for T2DA. The result shows **the superiority of utilizing world modeling to learn dynamics-aware task embeddings for text-decision alignment**.
>
> |Abaltion|**T2DA (CORRO)**|**T2DA**|
> |---|---|---|
> |Cheetah-Vel|-125.3±9.7|**-60.4±1.8**|
>
> [7] Towards an Information Theoretic Framework of Context-Based OMRL, NeurIPS 2024.
>
> ***
>
> **`Q4. Baseline choices.`**
>
> A4. **We include five baselines that cover three representative paradigms in tackling RL generalization**: CSRO, UNICORN, AD, BC-Z, and BAKU. We select CSRO and UNICORN to validate our motivation for leveraging language supervision over traditional OMRL methods.
>
> Both [8] and [9] learn hierarchical policies and fall into the paradigm of hierarchical RL. We did not compare T2DA to them due to not involving such a hierarchy. Instead, **we carefully select the most relevant and most representative baselines**: the classical BC-Z (citations: 649) [10] and the newer BAKU [11], to demonstrate our superiority in harnessing language knowledge.
>
> In summary, we justify that **we have included adequate and competitive baselines to verify the claim and contribution of our method**.
>
> [8] LISA: Learning Interpretable Skill Abstractions from Language, NeurIPS22.
>
> [9] Language Control Diffusion: Efficiency Scaling Through Space, Time and Tasks, ICLR24.
>
> [10] BC-Z: Zero-shot task generalization with robotic imitation learning, CoRL 2022.
>
> [11] BAKU: An efficient transformer for multi-task policy learning, NeurIPS 2024.
>
> ***
>
> **`Q5.  Report the “success rate” for Meta-World environment.`**
>
> A5. We selected returns rather than success rate as our evaluation metric for the Meta-World environment because the tasks vary significantly in difficulty. Some tasks readily achieve near 100% success rates, while others rarely succeed at all. **Returns provide a more nuanced and appropriate measurement for generalization problems across this spectrum of task difficulties**. Following your advice, we have reported the success rate of T2DA on Meta-World test tasks.
>
> |**Method**|faucet-open|drawer-close|reach-wall|handle-press-side|handle-pull-side|hand-insert|
> |---|---|---|---|---|---|---|
> |**T2DA**|0.11|0.82|0.13|0.38|0.21|0.34|
>
> ***
>
> **`Q6. The reason why T2DA outperforms other context-based OMRL approaches, and what fundamentally enables T2DA’s superior performance.`**
>
> A6. The superiority over traditional OMRL methods verifies our motivation of harnessing the representation power and knowledge transferability embodied in pre-trained language models. The Section 4.2 Ablation Study has thoroughly analyzed the respective contributions to T2DA. We compare T2DA to three ablations: w/o world, w/o align, and w/o text. In a progressive manner, we demonstrate that **T2DA’s performance comes from three aspects: 1) leveraging a wealth of knowledge from natural language, 2) contrastive knowledge alignment, and 3) capturing environment dynamics via the world model.** Also, the visualization insights in Figure 1 and Figure 3 again verify T2DA’s efficient comprehension of environment dynamics.
>
> ***
>
> **`Q7. Why T2DA maintains superior performance despite the use of lower-quality data, in Figure 5.`**
>
> A7. The absolute performance of any offline RL methods will be bottlenecked by the dataset quality. Both T2DA and baselines suffer a performance degradation when switching from Expert to Medium datasets. **T2DA’s superiority mainly comes from aligning text embeddings to comprehend environment dynamics**. The world model $p(s’,r|s,a)$ fully characterizes the environment, and remains invariant to behavior policies or collected datasets. Hence, **we can still distill the environment dynamics from lower-quality datasets using world modeling**. We evaluate T2DA on three types of datasets (Mixed, Expert, and Medium) to demonstrate our robustness to dataset quality.
>
> ***
>
> ### **`Summary Response`**
>
> Thank you for your valuable review comments, which help us gain more critical insights and further enhance our empirical evaluation. Also, we are grateful that **other reviewers** (we refer to qsSY as R1 and sYVw as R2) **make positive comments to our contributions, including the technical challenge** (’addresses a long-standing challenge in RL’ by R1 and ‘breaking through the traditional methods reliance on high-quality samples’ by R2), **the novelty** (’a well-engineered system…a unique conceptual advance’ by R1 and ‘a novel framework by…’ by R2), **and empirical evaluation** (’with excellent visual aids, e.g., t-SNE plots, training diagrams‘ by R1 and ‘comprehensive experimental validation’ by R2). We hope that our overall contributions have been adequately justified.
>
> We summarize that your main concerns may include the technical challenge, the novelty, and the baseline choice of our work. **We have extended a number of justifications and experimental analyses to address your concerns**. Please let us know if we have addressed your concerns. We are more than delighted to have further discussions and improve our manuscript. **If our response has addressed your concerns, we would be grateful if you could re-evaluate our work**.

---

> ### Comment · Reviewer_EUfm · 2025-08-02
>
> Thank you for your detailed response. However, I remain unconvinced and have summarized below the specific points I still find unclear. If I’ve misunderstood anything or any of my objections are misplaced, I would greatly appreciate your clarification.
>
> 1. I understand that T2DA addresses the reliance on high-quality samples or warm-up exploration by aligning task embeddings with language embeddings. However, I'm still not convinced that how this alignment actually taps into the pretrained text encoder's rich semantic knowledge, since the experiment text descriptions are so semantically minimal. For instance, in Cheetah-Vel and Ant-Dir the only differences between instructions is a numeric or angular parameter, e.g. "Please run at the target velocity of $v$", "Please walk toward the target direction of $\theta$".  Under these conditions, claims like “T2DA’s superiority mainly comes from aligning text embeddings to comprehend environment dynamics” or “leveraging a wealth of knowledge from natural language” seem unsubstantiated. The text encoder appears to serve merely as an interface for parameterized commands, with no clear evidence that leverage knowledge from the natural language.
>
> 2. I believe prior works extract per-timestep embeddings because, even when tasks differ, sub-trajectories with similar dynamics can still cluster together in embedding space. Moreover, there is nothing about transformer-based offline meta-RL methods [1,2] that prevents them from using an entire trajectory as a task prompt rather than a fixed-length segment. In fact, Meta-DT [1] trains its world model with the exact same loss as T2DA; the only difference is that Meta-DT ingests sub-trajectory instead of entire trajectory. There is no fundamental barrier preventing those methods from leveraging full-trajectory inputs, thus, it is unclear that encoding an entire trajectory as a single embedding as a significant extension of existing techniques.
>
> 3. The baseline comparisons don't seem fair. T2DA leverage a text encoder (e.g. BERT, T5, CLIP) and modern architectures like diffusion transformer, whereas the baselines (e.g. CSRO, UNICORN) seem to use MLP-based designs. It would strengthen the evaluation to include recent methods such as Prompt-DT [1], Meta-DT (decision transformer-based) [2], and MetaDiffuser (diffusion-based) [3].
>
> 4. It is hard to believe that simple language alignment alone could enable the agent to generalize to faucet-open (rotate faucet counter-clockwise) when it hasn’t been trained on that command, especially if the seen tasks are faucet-close )(rotate faucet clockwise) and door-unlock (unlock door by rotating counter-clockwise). For this, the model would have to infer that “close” in faucet-close is semantically analogous to “unlock” in door-unlock—both implying a counter-clockwise action—and then apply that action to a faucet, whose shape and mechanics differ substantially from a door handle. It’s unclear how such nuanced semantic and physical reasoning could emerge purely from command-level language alignment and contrastive learning. Moreover, I recommend that the authors also report the “success rate” of the baseline methods on the Meta-World environments, too.
>
> 5. In the “w/o text” ablation, if the text encoder is removed, where does the text input go during training, and how is any textual instruction used at inference time?
>
> [1] Prompting Decision Transformer for Few-Shot Policy Generalization, ICML 2022
>
> [2] Meta-DT: Offline Meta-RL as Conditional Sequence Modeling with World Model Disentanglement, NeurIPs 2024
>
> [3] MetaDiffuser: Diffusion Model as Conditional Planner for Offline Meta-RL, ICML 2023.

---

> > ### Author Response · Authors · 2025-08-03
> > **Further clarifications on the remaining minor issues**
> >
> > Thank you for your continued help and effort in improving our manuscript. **We are happy that the second round of comments only included some minor issues**, mainly about how our method works, comparison to transformer-based baselines, and auxiliary evaluation metrics. **We are grateful that our technical challenge and novelty have been adequately justified.** Next is our response to the remaining minor issues one-on-one.
> >
> > ***
> >
> > A1. **T2DA belongs to a language-conditioned policy learning paradigm, where a meta-policy learns to follow language instructions**. In simple environments like Point-Robot, the language instruction may seem like an interface for parameterized commands. However, in general scenarios like robotic manipulation, language instructions are usually highly abstract descriptions like “pull a handle up sideways” or “bypass wall and press button from top”. Our superiority comes from harnessing the representation power and knowledge transferability embodied in pre-trained language models. **The ablation study has already verified how this alignment actually taps into the pretrained text encoder's rich semantic knowledge**, i.e., T2DA’s performance comes from three aspects: 1) leveraging a wealth of knowledge from natural language, 2) contrastive knowledge alignment, and 3) capturing environment dynamics via the world model. Also, the visualization insights in Figure 1 and Figure 3 again verify T2DA’s efficient comprehension of environment dynamics.
> >
> > ***
> >
> > A2. In our case, encoding an entire trajectory as a single embedding **is more computationally efficient, more generalizable, and more applicable for text-decision alignment,** which is a significant extension of existing techniques. More importantly, our main contribution lies in **the first to align decision and language embeddings through world model dynamics,** where the VariBAD style is a specific way we utilized to distill the environment dynamics. It is very common for follow-up studies to adopt standard techniques to tackle new challenges in different domains. For example, MoSS adopts the VariBAD style and improves its task embedding by contrastive learning. MetaDiffuser adopts the VariBAD style and tackles the diffusion-based OMRL problem. Meta-DT adopts the VariBAD style to tackle the DT-based OMRL problem.
> >
> > ***
> >
> > A3. **We include five baselines that cover three representative paradigms in tackling RL generalization**: CSRO, UNICORN, AD, BC-Z, and BAKU. We select CSRO and UNICORN to validate our motivation for leveraging language supervision over traditional OMRL methods. We select BC-Z and BAKU (**built on strong transformer architectures**) to demonstrate our superiority in harnessing language knowledge. We have already included strong baselines built on modern architectures like transformers. We carefully and comprehensively include these representative baselines to justify our claim and overall contributions. **We did a careful baseline selection under the principle that the baselines should be the most relevant, high-quality, and influential.**  In the literature, there are endless baseline methods, and we can always pick out other methods in the literature in the next round.
> >
> > ***
> >
> > A4. Our experimental results have already verified that **simple language alignment alone could enable the agent to generalize to new tasks**. The ablation study has also verified that **such nuanced semantic and physical reasoning could emerge purely from command-level language alignment and contrastive learning**. We are also excited about the promising results achieved by harnessing the representation power and knowledge transferability embodied in pre-trained language models. For Meta-World, **the success rate and the received return are positively related**. Moreover, returns provide a more nuanced and appropriate measurement for generalization problems across this spectrum of task difficulties. That is why the received return is the most widely adopted evaluation metric in RL domains. Reporting the return can already verify our claim and overall contributions.
> >
> > ***
> >
> > A5. In the “w/o text” ablation, if the text encoder is removed, the method will degrade to the decision transformer or decision diffuser.

---

> ### Comment · Reviewer_EUfm · 2025-08-04
>
> Thank authors for their response. Overall, I agree that incorporating a text encoder enables the framework to avoid relying on high-quality samples or warm-up exploration; however, I remain unconvinced that the encoder’s semantic knowledge is actually leveraged to facilitate meta-RL. These concerns stem from the experimental design’s reliance on only simple text descriptions and from the unfair selection of baselines.
>
> 1. In the “w/o text” ablation, is the raw text still injected into the Decision Transformer (or Decision Diffuser)? If not, how does the model operate at inference time without any text input? And in this configuration, is the contrastive language–decision pre-training step also omitted? This should be clearly explained to help me to understand whether semantic knowledge from the pretrained text encoder is being leveraged.
>
> 2. The t-SNE plots in Figures 1 and 3 do not demonstrate any unique advantage from using a pretrained text encoder. They merely show that contrastive learning over decision embeddings can produce clustered embeddings. In the cheetah-velocity experiments, the text embeddings appear to be organized solely by the single “speed” parameter, which naturally sorts the points in order of velocity. This pattern could be replicated even without a pretrained text encoder by providing the raw speed scalar as input. This calling into question whether the text encoder is contributing any additional semantic structure under current setup. It would be more convincing to evaluate the model on highly abstract language instructions (as the author noted), so as to properly show the advantages of semantic encoding.
>
> 3. I appreciate the authors’ careful effort in selecting baselines, but the current comparisons don’t fully convince me of T2DA’s effectiveness. BAKU focuses on multi-task learning, and BC-Z targets language-conditioned policies. Since T2DA employs on a decision transformer and a diffuser, it would be more beneficial to include OMRL baselines that employ similar architectures (e.g. Meta-DT, Prompt-DT, Hyper-DT, MetaDiffuser). Additionally, providing a clear explanation of each model’s architecture and parameter size is essential to substantiate T2DA’s efficiency.
>
> 4. While I agree that RL is typically evaluated by cumulative reward, goal-conditioned benchmarks like MetaWorld, Franka Kitchen, and Maze2D are most often assessed by success rate (e.g., BC-Z, BAKU, Meta-DT). Showing that T2DA achieves superior success rates on these tasks would make authors' claims more convincing.

---

> > ### Author Response · Authors · 2025-08-04
> > **Another round of extended experiments for the remaining minor issues**
> >
> > Thank you for your time and continued effort. As you are still obsessed with the minor issues (i.e., how our method works, excessive baselines, and auxiliary evaluation metrics), **we conduct another round of extended elaboration and experiments to address them completely**.
> >
> > ***
> >
> > **`Q0. Whether the encoder’s semantic knowledge is actually leveraged to facilitate meta-RL.`**
> >
> > A0. Section 4.1 Main Results aim to demonstrate T2DA’s superiority in leveraging natural language supervision as a whole. Then, **Section 4.2 Ablation Study and Section 4.4 Visualization Insights aim to answer how the encoder’s semantic knowledge is actually leveraged to facilitate meta-RL**. First, comparing w/o text to w/o align highlights the necessity of leveraging a wealth of knowledge from natural language. Second, comparing w/o world to w/o align confirms the existence of a semantic gap between text and decision, whereas the gap can be effectively bridged by contrastive knowledge alignment. Finally, comparing T2DA to w/o world verifies that capturing environment dynamics via the world model can enable more precise and stable knowledge alignment between text and decision.
> >
> > **`Q1. The “w/o text” ablation.`**
> >
> > A1. As explained in the paper, the w/o text ablation completely removes the text encoder. Hence, the raw text is not injected into the DT or DD anymore, and there is no contrastive language-decision pre-training or the world modeling part.
> >
> > ***
> >
> > **`Q2. The t-SNE plots in Figures 1 and 3.`**
> >
> > A2. **The t-SNE plots in Figures 1 and 3 are not for demonstrating performance advantage, but for visualization insights** that verify T2DA’s efficient comprehension of environment dynamics. The performance superiority is verified by Section 4.1 Main Results, Section 4.2, and Section 4.3 Robustness Study.
> >
> > T2DA belongs to a language-conditioned policy learning paradigm, where a meta-policy learns to follow language instructions. In general scenarios like robotic manipulation, language instructions are usually highly abstract descriptions like “pull a handle up sideways” or “bypass wall and press button from top”. The reason why we chose Cheeta-Vel and Ant-Dir for visualization is that we can **map samples from different tasks to rainbow-colored points, where the task similarity can be easily observed from the color**. In this way, **we can gain intuitive visualization insights into how T2DA enables text embeddings to comprehend environment dynamics**. As appreciated by Reviewer qsSY, “with excellent visual aids (e.g., t-SNE plots, training diagrams) that intuitively convey the ideas behind decision-text alignment and dynamics-aware embedding”.
> >
> > ***
> >
> > **`Q3. Comparison to more OMRL baselines.`**
> >
> > A3. Thank you for your persistence in comparison to excessive OMRL baselines. Following your advice, we have included three additional OMRL baselines you mentioned: Prompt-DT, Meta-DT, and MetaDiffuser. We are very busy running experiments all day using four servers with a total of 22 GPUs. The result shows **T2DA’s consistent superiority over these excessive OMRL baselines.**
> >
> > | **Method** | Point-Robot | Cheetah-Vel | Ant-Dir |
> > | --- | --- | --- | --- |
> > | **Prompt-DT** | -14.3 $\pm$ 1.7 | -94.6 $\pm$ 8.1 | 895.2 $\pm$ 17.3 |
> > | **Meta-DT** | -8.4 $\pm$ 0.4 | -78.2 $\pm$ 5.3 | 922.1 $\pm$ 11.9 |
> > | **MetaDiffuser** | -11.7 $\pm$ 0.6 | -86.5 $\pm$ 4.8 | 858.5 $\pm$ 42.5 |
> > | **T2DA** | **-7.2 $\pm$ 0.1** | **-60.4 $\pm$ 1.8** | **970 $\pm$ 8.7** |
> >
> > ***
> >
> > **`Q4. Auxiliary evaluation metric of success rate.`**
> >
> > A4.  Thank you for your persistence in reporting the success rate in Meta-World. Following your advice, we have reported the success rate of T2DA and baselines on Meta-World test tasks. The result shows that **T2DA achieves superior success rates on these tasks, making our claims more convincing.** Note that AD needs 20+ hours to train and evaluate, we will include its result tomorrow.
> >
> > |**Method**|average|faucet-open|drawer-close | reach-wall | handle-press-side | handle-pull-side | hand-insert |
> > |---|---|---|---|---|---|---|---|
> > | **CSRO** | 0.19 | **0.15** | 0.42 | 0.09 | 0.24 | 0.11 | 0.12 |
> > | **UNICORN** | 0.20 | 0.06 | 0.53 | 0.07 | 0.17 | 0.13 | 0.21 |
> > | **BC-Z** | 0.23 | 0.04 | 0.73 | 0.03 | 0.21 | 0.18 | 0.16 |
> > | **BAKU** | 0.26 | 0.28 | 0.47 | 0.03 | 0.31 | 0.09 | **0.36** |
> > | **T2DA** | **0.33** | 0.11 | **0.82** | **0.13** | **0.38** | **0.21** | 0.34 |
> >
> > ***
> >
> > ### **`Summary Response`**
> >
> > Again, we greatly appreciate your continued engagement in reviewing our work and providing persistent concerns on minor issues. We have done a lot of hard work to provide extensive justifications and largely extended experiments in previous rounds. Also, we are grateful that Reviewer sYVw raised the score due to addressing his/her concerns, which makes our contributions and clarifications more solid. **We truly appreciate it if you could re-evaluate our work fairly, given our overall contributions and comprehensive responses in previous rounds.**

---

> > > ### Author Response · Authors · 2025-08-05
> > > **Full result of reporting the success rate in Meta-World**
> > >
> > > In the following table, we present the full result of reporting the success rate in Meta-World, including the AD baseline. Again, the result shows that **T2DA achieves superior success rates on these tasks, making our claims more convincing.**
> > >
> > > | **Method** | average | faucet-open | drawer-close | reach-wall | handle-press-side | handle-pull-side | hand-insert |
> > > | --- | --- | --- | --- | --- | --- | --- | --- |
> > > | **CSRO** | 0.19 | **0.15** | 0.42 | 0.09 | 0.24 | 0.11 | 0.12 |
> > > | **UNICORN** | 0.20 | 0.06 | 0.53 | 0.07 | 0.17 | 0.13 | 0.21 |
> > > | **BC-Z** | 0.23 | 0.04 | 0.73 | 0.03 | 0.21 | 0.18 | 0.16 |
> > > | **BAKU** | 0.26 | 0.28 | 0.47 | 0.03 | 0.31 | 0.09 | **0.36** |
> > > | **AD** | 0.13 | 0.08 | 0.26 | 0.10 | 0.14 | 0.07 | 0.13 |
> > > | **T2DA** | **0.33** | 0.11 | **0.82** | **0.13** | **0.38** | **0.21** | 0.34 |

---

> > > > ### Comment · Reviewer_EUfm · 2025-08-05
> > > >
> > > > Thank for the authors’ thorough rebuttal. I’m satisfied with authors' responses, and most of my concerns have been addressed. I hope these results are fully incorporated into the revised manuscript. Accordingly, I have increased my score.

---

> > > > > ### Author Response · Authors · 2025-08-05
> > > > > **Thank you!**
> > > > >
> > > > > Thank you very much for your constructive feedback and your time in helping us improve our work. We sincerely appreciate your increasing the score on our work! We will incorporate these extended clarifications and experiments into our revised manuscript.

---

### Official Review · Reviewer_sYVw · 2025-06-25

**Clarity:** 3
**Significance:** 3
**Originality:** 3
**Rating:** 4
**Confidence:** 4

**Summary:**

This paper proposes the Text-to-Decision Agent (T2DA) framework, which enhances the generalization capability of offline meta-reinforcement learning through natural language supervision and addresses the issue of traditional methods relying on high-quality samples. Its core approach involves encoding decision data into dynamics embeddings using a world model, aligning text with decision modalities via contrastive learning, and finally achieving text-to-decision generation through diffusion models and Transformers. Experiments demonstrate that T2DA outperforms baselines significantly on MuJoCo and Meta-World benchmarks, and exhibits robustness to data quality and text encoder selection.

**Questions:**

1. Eq. 3 calculates the cosine distance, the cosine distance is independent of the order of multiplication. Aren't these two similarities the same?
2. Eq. 4 uses contrastive learning, and $e^{sim(\tau_{i},l_{i})}$ in the denominator should be $e^{sim(\tau_{k},l_{i})}$? Otherwise, the non-diagonal elements of the similarity matrix in Figure 2b aren't used. Additionally, there is a minor error: the index of the second element on the diagonal of Figure 2b is incorrectly labeled.
3. The input of T2DA’s policy requires a text description, but under this setting, it resembles goal-based reinforcement learning or multi-task reinforcement learning rather than meta-reinforcement learning. For example, the text description of Point Robot directly includes the goal position, whereas in meta learning, the goal position cannot be known directly. Therefore, is it fair to compare T2DA with meta-learning algorithms like CSRO and UNICORN?
4. What is the impact on the results if the task is described in another kind of sentence?
5. Since a world model is used, T2DA should also be applicable to tasks with different dynamic functions. Could the authors supplement this part of the experiment?\
If the author answers my question, I am willing to increase my score.

**Ethical Concerns:**

["NO or VERY MINOR ethics concerns only"]

**Final Justification:**

The author's answer solved my problem

**Limitations:**

Yes

**Quality:**

3

**Strengths And Weaknesses:**

Strengths:\
The authors introduce natural language into offline meta-reinforcement learning, providing a novel framework by aligning text embeddings with environmental dynamics embeddings, thus breaking through the traditional methods' reliance on high-quality samples. Comprehensive experimental validation has demonstrated the necessity of each component of the method and the robustness of the algorithm.\
Weaknesses:\
The evaluation time of the method is relatively long. The description of the embeddings alignment module is unclear. The comparison with baseline algorithms is unfair. There is no description of the impact of textual changes.

---

> ### Author Rebuttal · Authors · 2025-07-30
>
> ***
>
> **`Q1. Difference from goal-based RL or multi-task RL.`**
>
> A1. Goal-based RL augments the observation with an additional goal to tackle multiple goals simultaneously, and the reward function is usually defined as a binary bonus of reaching the goal. In contrast, **T2DA belongs to a language-conditioned policy learning paradigm, where a meta-policy learns to follow language instructions**. In simple environments like Point-Robot, the language instruction can contain the goal, so it looks a bit like goal-based RL. However, in general scenarios like robotic manipulation, language instructions are usually highly abstract descriptions like “*pull a handle up sideways*” or “*bypass wall and press button from top*”. Apparently, it is hard to solve with goal-based RL.
>
> T2DA falls into the meta-RL category as we care about the zero-shot generalization to unseen test tasks. In contrast, multi-task learning focuses on maximizing performance on a set of training tasks.
>
> ***
>
> **`Q2. Fairness of comparing to meta-RL baselines.`**
>
> A2. As T2DA learns a language-conditioned meta-policy, we cannot ablate the language instructions. The key motivation of leveraging natural language supervision is to realize zero-shot generalization without inferring task beliefs from high-quality samples or warmup explorations (as meta-RL baselines do). **From both perspectives, T2DA uses language instructions to eliminate the need for few-shot samples, while meta-RL baselines need few-shot samples for task inference without language instructions**. Experiments in our manuscript have verified the significant superiority of T2DA’s paradigm over that of meta-RL baselines, which comes from harnessing the representation power and knowledge transferability embodied in pre-trained language models.
>
> Further, we modify the meta-RL baselines by adding the goal information to the observation, resembling a goal-conditioned setting. This ensures that the baselines can also access the goal information for a fairer comparison. **The result in the following table demonstrates T2DA’s consistent superiority over the goal-augmented baselines**. It further validates our motivation for leveraging natural language supervision. We will use this goal-augmented setting for meta-RL baselines in the revised paper.
>
> | **Method** | **CSRO (w/ goal)** | **UNICORN (w/ goal)** | **T2DA** |
> | --- | --- | --- | --- |
> | Cheetah-Vel | -84 $\pm$ 4.3 | -82.7 $\pm$ 9.2 | **-60.4 $\pm$ 1.8** |
> | Ant-Dir | 384.4 $\pm$ 30.1 | 427.1 $\pm$ 25.3 | **970.3 $\pm$ 8.7** |
>
> ***
>
> **`Q3. The impact on the results when the task is described by different textual descriptions?`**
>
> A3. Following your suggestion, we have conducted new experiments where tasks are captioned by two additional kinds of textual descriptions. We reconstructed task descriptions in two linguistic styles. **The first is a noisy style**: *"Um, there's this location I think... somewhere around {round(number=task[0], ndigits=2)}-ish in the x direction? And maybe {round(number=task[1], ndigits=2)} or so up?"*. **The second is a conversational style**: *"Could you move over to the spot that's roughly {round(number=task[0], ndigits=2)} units to the right and {round(number=task[1], ndigits=2)} units up? Thanks!”.*
>
> The result in the following table shows **T2DA’s consistent performance across different textual descriptions**. It also demonstrates the robustness of leveraging natural language supervision, as the powerful language representations can still capture task-relevant information and extrapolate the meta-level knowledge across tasks from different textual descriptions.
>
> | **Method** | **T2DA**  | **T2DA (Noisy)** | **T2DA (conversational)** |
> | --- | --- | --- | --- |
> | Point-Robot | $-7.2 \pm 0.1$ | $-7.6 \pm 0.1$ | $-8.0\pm 0.1$ |
>
> ***
>
> **`Q4. Experiments on tasks with different dynamic functions.`**
>
> A4. We agree with you that it is necessary to evaluate T2DA on both variations of the reward function and dynamics function, since these two functions constitute the whole world model. **Our experiments on Meta-World have already included tasks on different dynamic functions** $p(s’|s,a)$. While utilizing the same robotic arm, the interactions with different objects vary across tasks. These objects have different shapes, joints, and connectivity. For instance, the door has a revolute joint, while the drawer has a sliding joint, resulting in distinct dynamic functions.
>
> ***
>
> **`Q5. The evaluation time of the method is relatively long.`**
>
> A5. We train T2DA and baselines on one Nvidia RTX4080 GPU with the Intel Core i9-10900X CPU and 256G RAM. The following table shows the running time of T2DA and three types of baselines on the most complex environment Meta-World. The result demonstrates that **the computation time cost of T2DA is comparable to various baselines**.
>
> | **Method** | **Meta-RL baselines** | **AD baseline** | **Language-conditioned baselines** | **T2DA** |
> | --- | --- | --- | --- | --- |
> | Meta-World | 3h (CSRO) - 3.5h (UNICORN) | 22h | 1.5h (BAKU) - 3.5h (BC-Z) | 1.5h |
>
> ***
>
> **`Q6. The description of the embeddings alignment module is unclear. Two similarities in Eq. (3), and the use of non-diagonal elements in the similarity matrix of Eq. (4).`**
>
> A6. **The two similarities are the transpose of each other**, i.e., $sim(\tau,l)=sim(l,\tau)^T$. The reason we use two similarities is to highlight the difference between the text-to-decision score along the decision axis and the decision-to-text score along the text axis. Following CLIP, we use **a symmetric loss function** that contains cross-entropy losses along both the text and the decision axes for **contrastive alignment in both modalities**.
>
> **Non-diagonal elements.** Thank you for pointing out this typo. The correct decision-to-text score is calculated as $p(\tau_k)=\frac{e^{sim(\tau_k,l_k)}}{\sum_{i=1}^Ne^{sim(\tau_k,l_i)}}$ and the text-to-decision score is $p(l_k)=\frac{e^{sim(l_k,\tau_k)}}{\sum_{i=1}^Ne^{sim(l_k,\tau_i)}}$.
>
> Thank you for your professional comments on the contrastive alignment module.  We will improve the notations in this section to avoid potential misunderstandings, e.g., using a unified one similarity matrix.
>
> ***
>
> ### **`Summary Response`**
>
> Thank you for your valuable review comments, which help us gain more critical insights and further enhance our empirical evaluation. **We are honored to have your positive comments on our novelty** (”providing a novel framework”) **and the empirical evaluation** (”comprehensive experimental validation”). Also, we are grateful that **Reviewer qsSY is affirmative to our technical challenge** (”addresses a long-standing challenge in RL”), **the novelty** (”a well-engineered system…a unique conceptual advance”), **and the empirical evaluation** (”with excellent visual aids, e.g., t-SNE plots, training diagrams”). We hope that our contributions have been adequately justified.
>
> We summarize that your main concerns may include: 1) the difference from goal-based RL and fairness of comparing to meta-RL baselines, 2) T2DA’s performance with different textual task descriptions, and 3) experiments on tasks with different dynamics functions. **We have extended a number of justifications and experimental analyses to address your concerns**. Please let us know if we have addressed your concerns. We are more than delighted to have further discussions and improve our manuscript. **If our response has addressed your concerns, we would be grateful if you could re-evaluate our work**.

---

> ### Comment · Reviewer_sYVw · 2025-08-04
>
> The author's answer solved my problem. I will improve my score

---

> > ### Author Response · Authors · 2025-08-04
> > **Thank you!**
> >
> > Thank you very much for your constructive feedback and your time in helping us improve our work. We sincerely appreciate you raising the score on our work!

---

### Official Review · Reviewer_qsSY · 2025-06-26

**Clarity:** 3
**Significance:** 3
**Originality:** 3
**Rating:** 4
**Confidence:** 3

**Summary:**

This paper introduces Text-to-Decision Agent (T2DA), a novel framework for offline meta-reinforcement learning (meta-RL) that leverages natural language supervision to achieve zero-shot generalization across decision-making tasks. Traditional offline meta-RL approaches often rely on expensive task-specific signals (e.g., warmup explorations or expert data), whereas T2DA proposes to replace such supervision with textual task descriptions. Experiments on MuJoCo, Meta-World, and Point-Robot tasks demonstrate that T2DA significantly outperforms baselines from offline meta-RL, in-context RL, and language-conditioned imitation learning, particularly in challenging zero-shot settings.

**Questions:**

1. The test tasks are variants within the same environment families (e.g., new angles, velocities, or target locations). Have you explored cross-environment generalization, such as transferring from Point-Robot to Ant-Dir or from MuJoCo to Meta-World?

2. While you mention LoRA as a lightweight fine-tuning method for the text encoder, what is the performance or efficiency tradeoff compared to full fine-tuning?

3. In your experiments, the task descriptions are relatively structured and templated. How well does the method generalize to less structured, ambiguous, or noisy natural language instructions? Can you comment on robustness to varied linguistic phrasing or human-written descriptions?

**Ethical Concerns:**

["NO or VERY MINOR ethics concerns only"]

**Final Justification:**

I appreciate the authors' thoughtful responses and the revisions made in the manuscript. Overall, the revisions have significantly improved the clarity and strength of the paper. I maintain my positive assessment and recommend acceptance.

**Limitations:**

Yes.

**Paper Formatting Concerns:**

None.

**Quality:**

4

**Strengths And Weaknesses:**

Strengths:

1. The paper presents a well-engineered system that integrates offline meta-RL, world model pretraining, contrastive representation learning, and language-conditioned policy generation in a coherent and technically sound way.

2. The paper is clearly written, with excellent visual aids (e.g., t-SNE plots, training diagrams) that intuitively convey the ideas behind decision-text alignment and dynamics-aware embedding.

3. The proposed T2DA framework addresses a long-standing challenge in RL: enabling agents to generalize across tasks without task-specific adaptation or exploration.

4. The use of a dynamics-aware embedding space as an intermediate for text grounding is a unique conceptual advance.

Weakness:

1. While the framework is modular, it is unclear how robust it is to noisy or ambiguous natural language instructions, or how it performs in real-world environments beyond the benchmarks.

2. Some notations (e.g., $x_0(\tau)$ in diffusion model training) are dense and may require more intuitive explanation for non-specialists in generative models.

3. Some components build upon prior work (e.g., Prompt-DT, Decision Diffuser), so the originality mainly lies in their integration and adaptation rather than invention from scratch.

---

> ### Author Rebuttal · Authors · 2025-07-30
>
> ***
>
> **`Q1. How well does the method generalize to less structured, ambiguous, or noisy natural language instructions?`**
>
> A6. Thank you for your insighful question. Following your suggestion, we have conducted new experiments where tasks are captioned by two additional kinds of textual descriptions. We reconstructed task descriptions in two linguistic styles. **The first is a noisy style**: *"Um, there's this location I think... somewhere around {round(number=task[0], ndigits=2)}-ish in the x direction? And maybe {round(number=task[1], ndigits=2)} or so up?"*. **The second is a conversational style**: *"Could you move over to the spot that's roughly {round(number=task[0], ndigits=2)} units to the right and {round(number=task[1], ndigits=2)} units up? Thanks!”.*
>
> The following result shows **T2DA’s consistent performance across different textual descriptions**. It also demonstrates the robustness of leveraging natural language supervision, as the powerful language representations can still capture task-relevant information and extrapolate the meta-level knowledge across tasks from different textual descriptions.
>
> | **Method** | **T2DA**  | **T2DA (Noisy)** | **T2DA (conversational)** |
> | --- | --- | --- | --- |
> | Point-Robot | -7.2 $\pm$ 0.1 | -7.6 $\pm$ 0.1 | -8.0 $\pm$ 0.1 |
>
> ***
>
> **`Q2. Some components build upon prior work (e.g., prompt-DT and decision diffuser), rather than invention from scratch.`**
>
> A3. Foremost, **we address the key challenge in offline meta-RL** (OMRL) that usually requires either high-quality samples or warmup explorations to infer task beliefs for generalization. Instead of relying on these expensive supervision signals, we propose a novel solution using natural language supervision. Then, **we indeed address three fundamental challenges in learning generalist decision agents from natural language supervision**:
>
> - LLMs failing to capture any environment dynamics (we learn dynamics-aware decision embeddings);
> - Knowledge misalignment due to the semantic gap between text and decision (we align text embeddings to comprehend environment dynamics);
> - The necessity of scalable implementations (we develop scalable implementations of text-decision diffuser and text-to-decision transformer).
>
> **Our method is the first to align decision and language embeddings through world model dynamics**, which enables natural language understanding for decision task representation. Our primary contribution is a language-conditioned scalable framework for offline meta-RL. **The downstream policies in our framework are modular and can be implemented using various methods, even simple MLPs**. To promote scalability, we deploy T2DA on two mainstream generation architectures that hold the promise to train RL models at scale: autoregressive transformers and non-autoregressive diffusion models.
>
> ***
>
> **`Q3. What is the performance or efficiency tradeoff of LoRA fine-tuning compared to full fine-tuning?`**
>
> A4. Due to space constraints in the main paper, we've included a detailed analysis of the performance and efficiency tradeoffs between LoRA fine-tuning and full-parameter fine-tuning in Appendix F: "Analysis of Parameter-Efficient Fine-tuning.” The number of trainable parameters of LoRA is substantially smaller than full-parameter fine-tuning (less than 1%), enabling much more efficient computation and memory usage. Results show that with less than 1% parameter count, **T2DA using LoRA fine-tuning yields nearly identical performance compared to full-parameter fine-tuning, highlighting T2DA’s superior parameter efficiency without sacrificing effectiveness**.
>
> ***
>
> **`Q4. How does the framework perform in real-world environments?`**
>
> A4. While our current work focuses on tackling zero-shot generalization in offline meta-RL through text embedding alignment with environment dynamics, we have not yet conducted real-world robotic experiments. **The offline meta-RL community [1-3] primarily evaluates methods on simulation benchmarks such as MuJoCo and Meta-World** rather than physical robots. However, our framework shows promising potential for real-world applications - **the text understanding capabilities could help robots interpret natural language commands and generalize to new tasks**. We are willing to extend our method to real robotic systems in future work.
>
> [1] Offline Meta-RL with Advantage Weighting, ICML 2021.
>
> [2] Context Shift Reduction for Offline Meta-Reinforcement Learning, NeurIPS 2023.
>
> [3] Towards an information-theoretic framework of context-based offline meta-reinforcement learning, NeurIPS 2024.
>
> ***
>
> **`Q5. Have you explored cross-environment generalization, such as transferring from Point-Robot to Ant-Dir or from MuJoCo to Meta-World?`**
>
> A5. **Cross-environment generalization remains challenging in RL**, unlike natural language processing where semantically similar concepts share similar vector representations. The fundamental challenge lies in the state-action spaces: each environment has its unique structure and meaning, making direct transfer between different environments (like Point-Robot to Ant-Dir) impractical. While general RL aims to address cross-environment transfer [4], **our work focuses specifically on in-domain generalization, which is the standard of the offline meta-RL community** [1-3].
>
> The cross-environment generalization is a challenging yet significant problem to be investigated in RL domains, especially in the era of large foundation models. As mentioned in our limitations, an essential step is to implement on vast datasets with diversified domains, unleashing the scaling law with the diffusion or transformer architectures. We leave it as future work.
>
> [4] Towards General-Purpose Model-Free Reinforcement Learning, ICLR 2025.
>
> ***
>
> **`Q6. More intuitive explanation on diffusion model training for non-specialists in generative models.`**
>
> A6. We kept our explanation of the diffusion model concise due to paper space constraints. The diffusion model operates in two phases: In the forward process, noise is progressively added to a robot trajectory (states and actions) until it becomes random, analogous to blurring an image. In the reverse process, the model learns to reconstruct meaningful trajectories from noise by conditioning on task descriptions $\psi(l)$ and target returns $\hat{R}(\tau)$, similar to image denoising. During inference, the model generates appropriate action sequences by iteratively denoising random trajectories guided by natural language instructions. This enables zero-shot task execution through language conditioning without requiring additional training. **We will include more intuitive explanations in the main paper and more detailed descriptions in the appendix.**
>
> ***
>
> ### **`Summary Response`**
>
> Thank you for your valuable review comments, which have provided critical insights that helped strengthen our work. We are honored to have **your positive comments on our motivation and contribution** (”addresses a long-standing challenge…a unique conceptual advance…presents a well-engineered system”), **empirical evaluation** (”excellent visual aids, e.g., t-SNE plots, training diagrams”), **and the writing** (”the paper is clearly written”). Also, we are grateful that **Reviewer sYVw make positive comments on our novelty** (”providing a novel framework”) **and the empirical evaluation** (”comprehensive experimental validation”). We hope that our contributions can be adequately justified.
>
> We summarize that your main concerns may include: 1) robustness to ambiguous natural language instructions, 2) the choice of downstream components, and 3) discussions on real-world applications and cross-environment generalization capabilities. **We have extended a number of justifications and experimental analyses to address your concerns**. Please let us know if we have addressed your concerns. We are more than delighted to have further discussions and improve our manuscript. **If our response has addressed your concerns, we would be grateful if you could re-evaluate our work**.

---

### Comment · Area_Chair_ba7x · 2025-08-05
**Please provide your feedback on the authors' rebuttal.**

Dear reviewers,

For those who have not responded yet, please take a look at the authors’ rebuttal and update your final scores.

Best wishes,

AC

---

### Note · Authors · 2025-08-13

Dear Reviewers, ACs, SACs, and PCs,

We sincerely appreciate your help and effort in offering a good atmosphere for the discussion phase, where we are experiencing a positive communication process. We thank all reviewers (we refer to qsSY as R1, sYVw as R2, and EUfm as R3) for providing valuable comments to improve our work and for providing timely feedback when we successfully addressed their concerns. We have made a number of changes to address reviewers' suggestions and concerns. A short summary of the modifications is made as

- We conducted new experiments to **evaluate T2DA’s robustness to different kinds of textual descriptions**. (confirmed by R1, R2)
- We compared T2DA’s LoRA fine-tuning to full-parameter fine-tuning to **highlight T2DA’s superior parameter efficiency without sacrificing effectiveness**. (confirmed by R1)
- We highlighted **the superiority of T2DA’s paradigm over that of meta-RL baselines**, and further modified the meta-RL baselines by accessing the goal information to **ensure a fairer comparison**. (confirmed by R2)
- We presented the detailed running time to show that T2DA’s **computation time cost is comparable to various baselines**. (confirmed by R2)
- We highlighted **T2DA’s motivation** of leveraging natural language, **T2DA’s fundamental challenges** in supervising OMRL with natural language, and **T2DA’s key novelty** of being the first to align decision and language embeddings through world model dynamics. (confirmed by R3)
- We conducted new ablation experiments to **show T2DA’s superiority in learning task embeddings by world modeling**. (confirmed by R3)
- We included three new baselines and reported the success rate in Meta-World, to **demonstrate T2DA’s consistent superiority** over broader baselines evaluated by broader metrics. (confirmed by R3)

We are grateful that reviewers made positive comments on our overall contributions, such as the technical challenges, the novelty, and the empirical evaluation. Further, we are encouraged that **all reviewers gave positive feedback for our effective resolution of their concerns and expressed positive approval of our work**. We hope we were able to fully address all reviewers’ concerns to make our work more solid. Again, we truly appreciate your continued engagement in helping improve our work!

Best Regards,

The Authors

---

### Decision · Program_Chairs · 2025-09-17

**Decision:**

Accept (poster)

**Comment:**

The reviews (4,4,4) for this paper have been collected and discussed. There is a general consensus among the reviewers that the paper, in its current form, is suitable for publication. After careful consideration of the feedback and the paper itself, the recommendation is to accept this submission.